

# The "NorESM1-Happi" used for evaluating differences between a global warming of 1.5°C and 2°C, and the role of Arctic Amplification

Trond Iversen[1,2], Ingo Bethke[3], Jens B. Debernard[1], Lise S. Graff[1], Øyvind Seland[1], Mats Bentsen[3], Alf Kirkevåg[1], Camille Li[4], Dirk J. L. Olivié[1]

[1]Norwegian Meteorological Institute, P.O.Box 43, Blindern, 0313 Oslo, Norway

[2]Dep. of Geosciences, University of Oslo, P.O. Box 1047 Blindern, 0315 Oslo, Norway

[3]Uni Research Climate, Bjerknes Centre for Climate Research, P.O. Box 7810, 5020 Bergen, Norway

[4]Geophysical Institute, University of Bergen, Bjerknes Centre for Climate Research, P.O. Box 7803, 5020 Bergen, Norway

*Correspondence to:* T. Iversen (trond.iversen@met.no)

**Abstract.** The global NorESM1-M model that produced results for CMIP5 (http://cmip-pcmdi.llnl.gov/cmip5/index.html) has been slightly upgraded to NorESM1-Happi, and has been run with double resolution (~1° in the atmosphere and the land surface) to provide model simulations to address the differences between a 1.5°C and a 2.0°C warmer climate than the 1850 pre-industrial. As a part of the validation of temperature-targeted model simulations, the atmosphere and land models have been run fully coupled with deep ocean and sea-ice as an extension of the NorESM1-M which produced CMIP5-results. Selected results from a standard set of validation experiments are discussed: a 500-year 1850 pre-industrial control run, three runs for the historical period 1850-2005, three detection and attribution runs, and three future projection runs based on RCPs. NorESM1-Happi has a better representation of sea-ice, improved Northern Hemisphere (NH) extratropical cyclone and blocking activity, and a fair representation of the Madden-Julian oscillation. The amplitude of ENSO signals is reduced and is too small, although the frequency is improved. The strength of the AMOC is larger and probably too large. Modern era global near-surface temperatures and the cloudiness are considerably under-estimated, while the precipitation and the intensity of the hydrological cycle are over-estimated, although the atmospheric residence time of water-vapour appears satisfactory.

An ensemble of AMIP-type runs with prescribed SSTs and sea-ice from observations at present-day and a set of global CMIP5 models for a 1.5°C and a 2.0°C world (i.e. AMIP) has been provided by the model to a multi-model project (HAPPI, http://www.happimip.org/). This paper concentrates on the results from the NorESM1-Happi AMIP runs, which are compared to results from a slab-ocean



version of the model (NorESM1-HappiSO) designed to emulate the AMIP simulation allowing SST
and sea-ice to respond. The paper discusses the Arctic Amplification of the global change signal. The
slab-ocean results generally show stronger response than the AMIP results to a global change, such as
reduced NH extratropical cyclone activity, and different changes in the occurrence of blocking. A
considerable difference in the reduction of sea-ice in the Arctic between a 1.5°C and a 2.0°C world is
simulated. Ice-free summer conditions in the Arctic is estimated to be very rare for the 1.5°C case, but
to occur 40% of the time for the 2.0°C case. These results agree with some fully coupled models, but
need to be further confirmed.
**1    Introduction**
In *The Paris Agreement*, the parties to the United Nations Framework Convention on Climate Change
(UNFCCC) have established a long-term temperature goal for climate protection of "holding the
increase in the global average temperature to well below 2°C above pre-industrial levels and pursuing
efforts to limit the temperature increase to 1.5°C above pre-industrial levels, recognising that this
would significantly reduce the risks and impacts of climate change" (UNFCCC, 2015). This has
triggered considerable attention from climate modelling groups and researchers alike (e.g. Hulme,
2016; Peters, 2016; Rogelj and Knutti, 2016; Mitchell et al., 2016; Anderson and Nevins, 2016;
Boucher et al., 2016; Schleussner et al., 2016). For example, a special issue of the electronic journal
*Earth System Dynamics* focusing a 1.5°C or a 2°C global warming compared to pre-industrial
conditions is underway. A Special Report that will discuss the differences in climate and climate
impacts    between    a    "1.5°C    world"    and    a    "2.0°C    world"    is    under    development
(http://www.ipcc.ch/report/sr15/), and results in the present paper and many other scientific articles are
intended to contribute to that report.
In addressing differences in impacts of the ceilings 1.5 and 2°C in global warming, there are two basic
weaknesses of available climate projections from the Coupled Model Intercomparison Project (CMIP)
as reported in the assessment reports from the Intergovernmental Panel for Climate Change (IPCC).
The body of research assessing impacts under a 1.5°C world is small compared to higher emission
scenarios (James et al., 2017), and model simulations have been made on the basis of development
scenarios which give rise to a top of the model atmosphere (TOA) radiative forcing, rather than
selected temperature targets. New types of model calculations are therefore necessary to provide
support to the follow-up of The Paris Agreement (e.g. Mitchell et al., 2016). One example presented
by Sanderson, et al., (2017) is to develop a simple multi-parameter model which is designed to
emulate the model climate of a fully coupled climate model, and use this emulator to arrive at forcing
scenarios that produce developments towards a 1.5°C and a 2°C global warming in equilibrium.



We have decided to follow the approach presented in Mitchell et al. (2017) under the acronym *HAPPI*
(Half a degree additional warming, prognosis and projected impacts, http://www.happimip.org/). The
main multi-model ensemble experiments for HAPPI are of the AMIP-type (Atmospheric Model Inter-
comparison Project) for the present-day situation and for possible future situations which target the
agreed global temperature ceilings (Mitchell et al., 2017; and http://www.happimip.org/). These
experiments are based on prescribed sea-surface temperatures (SST) and sea-ice, although the
employed atmosphere and land model components are taken from a fully coupled ESM. Papers
dealing with the main multi-model results and analyses of differences in climate-related impacts
between a $1.5^{\circ}C$ and a $2.0^{\circ}C$ warmer world (relative to the 1850 pre-industrial climate) from HAPPI,
are submitted to the mentioned special collection of papers in Earth System Dynamics (*The Earth*
*System at a global warming of 1.5°C and 2.0°C*).
The present paper documents a selection of major properties of the coupled Norwegian Earth System
Model dedicated for HAPPI (*NorESM1-Happi*). The model is a modified version of the NorESM1-M
(Bentsen et al., 2013; Iversen et al., 2013, Kirkevåg et al., 2013) used for the fifth cycle of CMIP
(CMIP5) and referred to in the fifth assessment report of IPCC (IPCC, 2013). By October 2017,
NorESM1-M has been referred to in almost 500 CMIP5-related publications (https://cmip-
publications.llnl.gov/search?type=model&option=NorESM1-M). The model is also equipped with a
thermodynamic slab ocean (SO) model (*NorESM1-HappiSO*). Throughout the paper, NorESM1 refers
to either of the two model versions, while the extensions -M and -Happi are used to specifically denote
one of them. Since Bentsen et al. (2013) and Iversen et al. (2013) presented, in great detail, the
NorESM1-M, this paper emphasizes selected properties of NorESM1-Happi and changes since
NorESM1-M.
The role of Arctic Amplification (AA) for a given level of global warming (Arrhenius, 1896; Manabe
and Stouffer, 1980, Holland and Bitz, 2003, Feldl et al., 2017) is relevant for the consequences of the
Paris agreement. The relevance of AA is first of all due to in-situ changes in the sea-ice and snow-
cover. In the next instance it can be relevant for the potential trigger of Arctic-specific irreversible
feedbacks, as well as the possible (but still controversial) changes in quasi-persistent mid-latitude
weather patterns due to reduced meridional temperature contrasts (Francis and Vavrus, 2012; Screen
and Simmonds, 2013; Cohen et al., 2014; Barnes and Polvani, 2015). Although the amplitude and
pattern of AA varies between models, it is nevertheless a robust response to a positive radiative
forcing predominantly driven by a positive regional lapse-rate feedback (negative at lower latitudes)
and a positive albedo feedback due to reduced sea-ice and snow-cover (Winton, 2006; Pithan and
Mauritsen, 2014). Even for the remote and regionally localized forcing caused by reduced European
sulphate aerosols since the 1980s, the AA has been calculated with a strong response (Acosta Navarro
et al., 2016).



We use the slab-ocean model (NorESM1-HappSO) to predominantly assess how the AA may differ
between equilibrated 1.5$^{o}$C and 2$^{o}$C global warming levels since the 1850 pre-industrial level. A
weakness of the AMIP-type approach of HAPPI is the enforcement of prescribed SST and sea-ice
fields. This leads to a pattern of AA which is basically pre-defined and does not belong to a natural
realization of the model-climate's SST and sea-ice. Representing the upper ocean as thermodynamic
slab with a fully dynamic sea-ice description attached to it, is not an ideal solution, but when it is
properly calibrated, it includes more degrees of freedom for the model when responding and
equilibrating with the forcing of the system. The reliability of the SO model results is higher if their
quality is approximately the same or better than the fully prescribed experiments.
The calibration of the SO model is tuned to fit the driving conditions for the HAPPI experiments
performed with NorESM1-Happi, but without prescribing the SST and sea-ice. The SO experiments
thus allow explicit feedback processes involving the SSTs and the sea-ice, but without calculation of
ocean currents. Further comparisons with results from the NorESM1-Happi coupled to the deep ocean
are done as well, although in that case without sufficient accounting for model uncertainties and
internal variability.
This paper first describes the ingredients of the NorESM1-Happi model in section 2 emphasizing the
changes since NorESM1-M, and the slab ocean version NorESM1-HappiSO in section 3. Section 4
presents and discusses the properties of NorESM1-Happi as a coupled climate model, while some
aspects of the 1.5$^{o}$C and 2.0$^{o}$C global temperature targets are discussed in Chapter 5, emphasizing
Arctic amplification, extratropical cyclone activity and blocking, and the fate of Arctic sea-ice. Some
conclusions are given towards the end.

## 2   The NorESM1-Happi model

This section reviews selected features of NorESM1, emphasizing own developed model processes and
differences between the model versions -M and -Happi. More complete descriptions of NorESM1-M,
and thus also most of NorESM1-Happi, are found in section 2 of Bentsen et al. (2013). NorESM1 is
based on the fourth version of the Community Climate System Model (CCSM4) developed in the
Community Earth System Model project at the US National Center for Atmospheric Research
(NCAR) in collaboration with many partners (Gent et al., 2011). Explicit description of the interactive
carbon cycle is not included, and the experiments employ prescribed concentrations of greenhouse
gases (GHG).
Changes in NorESM1-Happi compared to NorESM1-M were mainly developed in the project
ACCESS (Arctic Climate Change, Economy and Society) financed under the 7$^{th}$ Framework
Programme of the European Union (http://www.access-eu.org/). The modifications included double





horizontal resolution in the atmosphere and land model and a different tuning of the aging, and thus
albedo, of snow accumulated on sea-ice (Seland and Debernard, 2014). More recently an error in the
aerosol scheme (Kirkevåg et al., 2013) was found and rectified, resulting in faster condensation of
secondary gaseous matter on pre-existing particles.
## 2.1    The Atmosphere
NorESM1's atmospheric component is based on the original CAM4 publicly released in April 2010
(Neale et al., 2010 and 2013). The finite volume dynamical core for transport calculations (Rasch et
al., 2006) is used. In NorESM1-Happi the horizontal resolution is 0.95º latitude by 1.25º longitude (in
short: 1º resolution), which is half the mesh-width used in NorESM1-M. In the vertical direction, 26
levels are used with a hybrid sigma-pressure co-ordinate and model top at 2.194 hPa.
In NorESM1 the atmospheric model (CAM4-Oslo) is extended by calculating online the lifecycles of a
range of natural and anthropogenic aerosol components from emissions and physico-chemical
processing in air and cloud droplets. The only prescribed aerosol concentrations are stratospheric
sulphate from explosive volcanoes in the historical period. Other aerosol components are
calculated from emissions. Correct modelling of forcing of anthropogenic aerosols depends
on the representation of natural background aerosols and the associated cloud droplet
properties (e.g. Hoose et al., 2009). Parameterization of aerosol interactions with the model's
schemes for radiation and warm cloud microphysics enable the calculation of direct and indirect
aerosol effects on climate (Kirkevåg et al., 2013). The model employs a prognostic calculation of
cloud droplet numbers using the Abdul-Razzak and Ghan (2000) scheme for condensation nuclei
activation, allowing for competition effects between aerosols of different hygroscopic property and
size.
The upper air burdens of aerosols and other atmospheric constituents exposed to precipitation
scavenging are sensitive to processes taking place in deep convective clouds (Seland et al., 2008). In
NorESM1 deep moist convection is parameterized as in CCSM4 by the scheme of Zhang and
McFarlane (1995), extended with plume dilution and convective momentum transport (Richter and
Rasch, 2008; Neale et al., 2008).  Both Samset et al. (2013) and Allen and Landuyt (2014) indicated
that NorESM1-M had too high upper air concentrations of black carbon (BC) aerosols. This could
cause overestimated absorption of solar radiation, suppressed upper-level cloudiness, and exaggerated
static stability.
The physical properties of BC differ from other anthropogenic aerosols by its occurrence in
hydrophilic form after pyrolysis and inefficient combustion of fossil fuel. By mixing with hygroscopic
matter, BC aerosols become hydrophilic, and the efficiency of this BC aging reduces the ability of BC



to survive vertical transport in deep convective clouds (Allen and Landuyt, 2014). The increased
efficiency of aerosol condensation in NorESM1-Happi, enhanced the scavenging efficiency of BC
compared to NorESM1-M, and Fig. 1 shows considerably reduced atmospheric burdens of BC, in
particular in the middle layers of the free troposphere. Nevertheless, to the extent that the observed
vertical profile of BC during the HIPPO-campaign (Schwartz et al., 2013) is representative for the
vertical distribution of BC in general, the model still mixes the BC too high up, and the transport and
scavenging in deep convective clouds probably needs to be improved. Table 1 summarizes the global
budget for the aerosol particles and their precursors for NorESM1-Happi and NorESM1-M. All but
BC have minor changes in atmospheric burdens and residence times, while the reduction for BC is
substantial. It should be mentioned that a comprehensive discussion of the aerosols in NorESM is
underway in another publication.

## 2.2 The Ocean

For NorESM1, the ocean model in CCSM4 is replaced by an elaborated version of the Miami
Isopycnic Community Ocean Model (MICOM) adapted for multi-century simulations in coupled
mode (Assmann et al., 2010; Otterå et al., 2010). The extensions of the ocean model since the original
MICOM are summarized by Bentsen et al. (2013) and include improved parameterization of diapycnal
mixing, thickness and isopycnal eddy diffusion, and the mixed layer depth. A grid with 1.125º
resolution along the equator is used with the Northern Hemisphere (NH) grid singularity located over
Greenland. The grid is a standard (gx1v6) provided by CCSM4, and the same ocean mask is used. The
bathymetry is created by averaging the depths of a high resolution data set (S2004; Marks and Smith,
2006) belonging to each ocean grid cell, and editing of the bathymetry is limited to setting key sills
and channels to their actual depths. A total of 53 model layers are used with layer reference potential
densities in the range 28.202–37.800 kg m$^{-3}$. Originally, MICOM has a single bulk surface mixed
layer. In NorESM1 the mixed layer is described by two model layers with freely evolving density. The
two layers are equally thick when the mixed layer is shallower than 20m, while the uppermost layer is
limited to 10m when the mixed layer is deeper than 20m.

In connection with the EU ACCESS project, external inertia-gravity waves were damped in shallow
regions in order to remove spurious oceanic variability in high latitude shelf regions. This damping is
kept in NorESM1-Happi.

## 2.3 The Sea-Ice

*The sea ice model* in NorESM is the original CICE4 version used in CCSM4 (Gent et al., 2011;
Holland et al., 2012). The model employs the delta-Eddington short-wave radiation transfer (Briegleb



and Light, 2007), and parameterization of melt ponds and aerosols in snow and ice. The sea ice
component is configured on the same grid as the ocean component.
In NorESM1-Happi, wet snow albedo on sea-ice was decreased compared to NorESM1-M by
increasing the wet snow grain size and by allowing a more rapid metamorphosis from dry to wet snow.
This affects the Arctic sea-ice more than the Antarctic, since the latter is less frequently influenced by
mild and humid air.
As a result of the sea-ice albedo change and the reduced wave-induced oceanic variability along high-
latitude shelves, the large-scale distribution of sea-ice thickness across the Arctic was considerably
improved compared to NorESM1-M (Seland and Debernard, 2014; see also Fig. 11 below).
**2.4 The Land**
NorESM1 employs the original version 4 of the Community Land Model (CLM4) (Oleson et al.,
2010; Lawrence et al., 2011) of CCSM4, which included the SNow, ICe, and Aerosol Radiative model
(SNICAR; Flanner and Zender, 2006). The surface albedo and the vertical absorption profile depend
on solar zenith angle, albedo of the underlying snow, mass concentrations of atmospheric-deposited
aerosols, and ice effective grain size simulated with a separate snow aging routine. Atmospheric-
deposited aerosol components treated by SNICAR in NorESM1-Happi are black carbon and mineral
dust. In the experiments, the carbon–nitrogen (CN) cycle option of CLM4 is enabled (Thornton et al.,
2007; Gent et al., 2011), although their fluxes are diagnostically determined and do not influence other
model components. CLM4 employs the same horizontal grid as CAM4-Oslo, except for the river
transport model which is configured on its own grid with a horizontal resolution of 0.5°.
**2.5 The Coupler**
*CPL7* was developed specifically as a coupler for CCSM4 (Craig et al., 2012) and it is used in
NorESM1 without changes. It controls the exchange of information between model components and
the execution of the coupled system, by organizing the components of the coupled model into a single
executable, and issuing calls to initialization, run, and finalization routines for each model component.
The components can be configured to run sequentially, concurrently, or as a combination, thus
enabling optimal use of the hardware resources.
In NorESM1, fields and fluxes are exchanged between the components half-hourly, except for the
ocean components that are coupled once per day. The land and ice components are responsible for
computing the atmosphere/land and atmosphere/ice fluxes, respectively, while the coupler computes
the atmosphere/ocean fluxes every half hour, providing the instantaneous fluxes to the atmosphere and
daily mean fluxes to the ocean component.





## 3   Emulating the oceanic response with a Slab Ocean (SO)

A slab ocean model (SO) has been set up for a specific purpose related to the design of the multi-model experiments of the HAPPI-project (Mitchell et al., 2017; http://www.happimip.org/). Those experiments are of the AMIP type, in which any change in the sea-surface temperature (SST) and sea-ice cover are prescribed in combination with changes in the major elements producing radiative forcing in the representative concentration pathways RCP2.6 and RCP4.5. Combinations of these elements are constructed based on the CMIP5 multi-model ensemble in order to target the two imagined future states defined by increase in global surface air temperature (SAT) above the 1850 pre-industrial level: 1.5ºC and 2.0ºC. Even though the prescribed changes to future states are based on fields from fully coupled model runs, the feedbacks behind amplification of climate change signals will not be present in the HAPPI experiments. With NorESM1-HappiSO we intend to investigate if this calculated amplification may depend on feedbacks that involve changes in Arctic sea-ice and SST which are not present in the multi-model HAPPI experiments.

A slab ocean model does not calculate ocean circulation and associated fluxes, but treats the upper ocean mixed layer as a single layer which buffers heat-fluxes through the ocean surface, i.e. a thermodynamic "slab" governed by the equation

$$\rho_0 c_0 h_{mix} \frac{\partial SST}{\partial t} = F_{net} - Q_f \tag{1}$$

where the thickness $h_{mix}$ of the slab varies in space but not in time, $\rho_0$, and $c_0$, respectively, are the density and heat capacity of the mixed layer sea-water, $SST$ is the sea-surface temperature which in this connection equals the mean, mixed-layer temperature. $F_{net}$ is the net input of heat through the ocean surface from the atmosphere and sea-ice, and $Q_f$ is the net divergence of heat not accounted for by the explicit processes needed to maintain a stable climate with a predefined geographical distribution of SST.

The realism of the SO model climate depends on how $Q_f$ is prescribed. In Bitz et al (2012), $Q_f$ is calculated using $h_{mix}$, $SST$, and $F_{net}$ from a fully-coupled stable control simulation. Both $h_{mix}$ and $SST$ should represent an assumed well-mixed layer in the vertical. With an annual mean (but still spatially variable) mixed-layer thickness, it is quite straightforward to obtain balance with the annual cycle of heat (Bitz et al., 2012). This method would give a SO model mean SST distribution very similar to, and consistent with, the climate of the fully coupled model when external forcing is unchanged. Here, this method has been used when estimating the equilibrium climate sensitivity (ECS) for runs with abrupt $CO_2$ doubling ($\Delta T_{eq} = 3.31\ K$) and $CO_2$ quadrupling ($\Delta T_{eq4} = 6.74\ K$), giving an average $\Delta T_{eq} = 3.34\ K$ for doubling of the atmospheric $CO_2$ –concentrations, Table 5). The

$Q_f$ used in these experiments was diagnosed from the 1850 fully coupled pre-industrial control run
with NorESM1-Happi (Section 4.3), and kept constant in the different SO runs.
### 3.1    Calibration of NorESM1-HappiSO experiments
One drawback with this method for quantifying $Q_f$, is that biases in SST and the mean climate from
the fully coupled model are kept in the SO model, which makes comparison with the AMIP
experiments for HAPPI difficult. Therefore, as an alternative, we also use a restoring method similar
to Williams et al. (2001) and Knutson (2003), where a separate calibration run of the SO is done with
an additional restoring term $(SST - SST_{ext})/\tau$ added to the right-hand side of (1). $SST_{ext}$ is an
externally imposed SST-field valid for some specific period (observation- or model-based) with $\tau$ as a
prescribed time-scale for adjustment. After this run, the new $Q_f$ is defined by adding the monthly
climatology of the restoring flux to the $Q_f$ used in the calibration run. Then, when used in a free SO
run (without restoring), the new $Q_f$ ensures a modelled SST climate close to the $SST_{ext}$ fields imposed
during the calibration. We have kept the sea-ice model free without any restoring or constraints to
observed fields during the calibration. This increases the realism of the ice-ocean heat fluxes going
into $F_{net}$, and ensures consistent changes in sea ice mass and energy. As in Bitz et al. (2012) the sea-
ice in the SO set-up employs the full CICE4 dynamic and thermodynamic model, which is the same as
used in the fully coupled NorESM1-M and NorESM1-Happi. However, some tuning of snow albedo
over sea ice have been done to increase the realism of sea ice extent under present day conditions
when using the restoring method for specifying $Q_f$.
In the present case, the purpose of the SO-model is to emulate regional patterns of the climate
response given a targeted ceiling for the global reference height temperature change relative to the pre-
industrial 1850 climate, taking into account the observed and analysed climate at present-day (2006-
2015). The experiments with NorESM1-HappiSO are designed to be directly comparable with the
AMIP-type runs with NorESM1-Happi for HAPPI, in which the SST and sea-ice are prescribed (see
Ch. 5). Three different calibrations of $Q_f$ are therefore performed using the restoring method. For
present-day (PD) we use 10-year averaged SSTs determined by the observationally based *Operational*
*Sea Surface Temperature and Sea Ice Analysis* (OSTIA) for the decade 2006-2015 (Donlon et al.,
2012). In practice, this calibration also removes biases. For the future 1.5°C and 2.0°C climate scenario
states, we determine the $Q_f$ which adjusts to the 10-year average SSTs used as input to the AMIP-type
HAPPI experiments. These were obtained by adding SST increments from the multi-model CMIP5
data to the OSTIA PD SST field. The different $Q_f$-fields thus emulate the effects of oceanic
circulation changes on the heat flux divergence in the well mixed layer.





The $Q_f$-fields are determined for each month of the year, and the values used in the slab ocean model
at a given grid-point and a given time is determined by linear interpolation between the former and the
next monthly value. The same $Q_f$-fields are used every year. Fig. 2 shows annual averages for the
present-day determined by the protocol for the HAPPI AMIP experiments (2006-2015) together with
the increments for the 1.5°C and the 2.0°C warmer world. In addition, we use the same $CO_2$-levels,
aerosols and precursor emissions, and other active forcing-agents as in the HAPPI AMIP experiments.
The $Q_f$ for present-day (PD), which includes bias corrections, is dominated by large negative values
(hence SST increase) along the major currents in the North Pacific, North Atlantic, Southern Indian
Ocean, and the Atlantic sector or the Arctic. Positive values are mainly seen along the equator and in
some coastal upwelling zones. The increment patterns appear largely independent on the level of the
warming, with positive values (decreasing SST) over the Labrador Current, and negative values
(increasing SST) mainly in the tropics and north-west Pacific.
## 4    Coupled Pre-industrial Control, Historical, and Projection Runs
Throughout this paper, we use "piControl" to identify a 500-year control simulation with constant
external forcing prescribed for 1850 conditions. Aerosols and precursor emissions are as in Lamarque
et al. (2010). Emissions of sea-salt are calculated on-line according to surface wind speed and SST.
Concentrations of greenhouse gases (GHG) are consistent with preindustrial conditions in accordance
with CMIP5 (http://cmip-pcmdi.llnl.gov/cmip5/forcing.html). The incoming solar flux at the model
top is constant at 1360.9 W m$^{-2}$ and the $CO_2$ mixing ratio is constant at 284.7 ppm.
Before the piControl, the 1° atmospheric and land resolution ACCESS version of the model (Seland
and Debernard, 2014) was spun up over 300 years starting from year 600 of the NorESM1-M spin-up
with 2° resolution atmosphere and land (Bentsen et al., 2013). The ocean and sea-ice were in both
cases run with 1° resolution. At year 900, the bug-fix for condensation in the aerosol-scheme was
introduced (see section 3.1), and the model was run for 500 years. These latter 500 years are
considered the piControl run for NorESM1-Happi, even though some adjustments over the first years
due to the aerosol fix should be expected. The aerosol fix is mainly affecting the upper air BC
concentrations (Table 1 and Fig. 1) with minor impacts on surface temperatures, surface energy fluxes,
and multi-decadal variability associated with the deep oceans (Sand et al., 2015; Stjern et al., 2017).
Three independent ensemble members to simulate the historical period from 1850 to 2015 were
branched off from the piControl at years 920 ("Hist1"), 950 ("Hist2") and 980 ("Hist3"). From 1850 to
2005 natural variations of solar radiation (Lean et al., 2005; Wang et al., 2005) and stratospheric
sulphate aerosol concentrations from explosive volcanoes (Ammann et al., 2003), as well as
anthropogenic GHG concentrations, aerosol emissions, and land-cover changes were prescribed using





the CMIP5 forcing-data (http://cmip-pcmdi.llnl.gov/cmip5/forcing.html). Forcing detection and
attribution experiments were run for the Hist1 experiment for "GHG only" (GHG=greenhouse gases),
"Aerosol only", and "Natural forcing only".
From 2005 onwards, the representative concentration pathway (RCP) scenarios (van Vuuren et al.,
2011) were the basis for a prolongation of Hist1 for climate projections until 2100. NorESM1-Happi
has run projections for RCP2.6, RCP4.5, and RCP8.5. The historical ensemble members were all
extended to 2015 using RCP8.5 for the years 2006-15.

## 8  4.1  Climate stability and present-day characteristics

NorESM1-Happi is a version of NorESM1-M with relatively minor updates. NorESM1-M is
thoroughly documented through CMIP5. The most radical change is the double horizontal resolution
in the atmosphere and land models. Many of the properties of NorESM1-Happi are therefore in reality
well documented. This paper presents selected major features only.
Fig. 3 shows the variation over the 500-year control period of selected key variables for the long-term
stability of the global climate, and Table 2 compares average values from Hist1 for 1976-2005 with
relevant observation and re-analyses. These can be compared with Fig. 2 and Table 1 in Bentsen et al.
(2013). Linear trends are estimated by linear regression of annual averages, and the statistical
significance of trends investigated by a t-test after adjusting the degree of freedom to account for
autocorrelation (Bretherton et al., 1999). A value of $p < 0.05$ is assumed to indicate that a non-zero
linear trend is statistically significant.
In the following we compare numbers from NorESM1-Happi with *numbers from NorESM1-M given*
*in brackets*, unless otherwise indicated. The global 500-year mean net radiation at the top of the model
atmosphere (TOA) is -0.042 $Wm^{-2}$, well below the desired range of $\pm0.1$ W $m^{-2}$. The linear 500-year
trend is +0.001 W $m^{-2}$, and is not statistically significant. The NorESM1-M value was +0.043 W $m^{-2}$
(the value +0.086 W $m^{-2}$ reported in Bentsen et al. (2013) was inaccurately calculated). The flux
imbalance at the TOA causes a small sustained cooling of the earth system, as opposed to a warming
in NorESM1-M. The corresponding net heat flux into the ocean is +0.004 W $m^{-2}$ (+0.122 W $m^{-2}$) with
a statistically insignificant linear trend of +0.004 W $m^{-2}$ (-0.019 W $m^{-2}$). This tiny heat-flux causes the
global mean ocean temperature of 3.78$^o$C (3.81$^o$C) to increase by +0.008 K (+0.126 K) over 500 years,
which is statistically significant. A slow but statistically significant freshening of the ocean water
masses occurs over the 500 years, as the average ocean salinity of 34.72 g $kg^{-1}$ (in both models) has a
negative change of 3.20 $10^{-4}$ g $kg^{-1}$ (3.14 $10^{-4}$ g $kg^{-1}$). Fig. 4 shows that the salinity probably is
generally underestimated in the upper km of the ocean and overestimated below, while the trend is
hardly visible, as opposed to the results for NorESM1-M (Fig. 3 in Bentsen et al., 2013). The sea-



surface salinity (SSS) has an average global value of 34.57 g kg$^{-1}$ (35.49 g kg$^{-1}$) with a small and
statistically significant positive (p=0.03) 500-year change of +0.005 g kg$^{-1}$.
In general, the piControl of NorESM1-Happi appears to be more stable and with smaller deviations
from the World Ocean Atlas of 2009 than NorESM1-M (compare Fig. 4 with Fig. 3 in Bentsen et al.,
2013). However, this long-term stability may to some extent be a consequence of an even stronger
Atlantic Meridional Overturning Circulation (AMOC) than in NorESM1-M. The 500-year average is
32.4 Sv (30.8 Sv) with a considerable and statistically significant positive 500-year increase of 1.0 Sv
(-0.6 Sv). As discussed in Iversen et al. (2013), a strong AMOC efficiently mixes heat into the deep
ocean, leaving less for surface heating and evaporation. The transport through the Drake Passage is
also larger in NorESM1-Happi than in NorESM1-M in the 1850 control run (135 Sv vs. 130 Sv), but
both models produce a statistically significant 500-year change (-6.98 Sv vs. -6.29 Sv).
The heat lost through the TOA and into the ocean from remaining parts of the climate system, i.e. the
atmosphere, the land surface, and the sea-ice, is 0.045 W m$^{-2}$ (0.043 W m$^{-2}$ for NorESM1-M), and is
comparable to the 1 degree version of CCSM4 which had a stronger negative TOA radiative heat
balance and also a negative heat flux at the ocean surface. The pre-industrial global near surface air
temperature of NorESM1-Happi is 12.74$^{o}$C with a statistically significant (p=0.04) cooling of 0.032 K
over the 500 years. In comparison, NorESM1-M produced a 0.39 K higher pre-industrial near-surface
temperature with a statistically significant 500-year warming of 0.037 K. The average global sea-
surface temperature (SST) of 17.37$^{o}$C (17.68$^{o}$C) has a statistically insignificant 500-year decrease of
0.021 K (+0.03 K increase).
The pre-industrial temperatures are by construction supposed to be valid for a postulated stable climate
at 1850, and cannot adequately be compared to observations. Instead, the climate simulated with the
Hist-experiments for a recent 30-year period, e.g. 1976-2005, should be comparable to observationally
based data, see Tables 2, 3, 4, and column no. 3 from the right in Table 6. Both NorESM1 versions
produce a cold bias averaged over the global continents, but the negative bias is 0.52 K larger in
NorESM1-Happi (Table 3). Similarly, the negative SST bias is 0.31 K larger in NorESM1-Happi. This
larger bias is likely associated with higher horizontal resolution, in agreement with the experience
from the CCSM4-model (Gent et al., 2011). The cloud and precipitation parameterizations in CAM4
(the atmospheric model in CCSM4) and in CAM4-Oslo (used with NorESM1) tend to reduce the
cloud cover as grid resolution increases, and the ground surface loses more heat in the 1 degree
version than in the 2 degree version. This is corroborated by the numbers in Table 2 for both
NorESM1 versions, even though the net cloud radiative forcing itself is slightly less negative in
NorESM1-Happi, contributing to a warming compared to NorESM1-M. Other differences between the
models may also contribute, such as reduced absorption of solar radiation in clear air due to smaller
atmospheric burdens of BC, but this effect is masked by the effects of reduced cloud cover in



NorESM1-Happi. More short-wave (SW) radiation is absorbed by the ground surface, which
predominantly increase the latent heat flux by 2 W m$^{-2}$ (Table 2). Furthermore, while the net radiative
imbalance at the TOA (ca. +0.5 W m$^{-2}$, see Fig. 5) is similar in the two model versions, the larger TOA
influx of SW radiation in NorESM1-Happi is almost entirely compensated by increased TOA outgoing
LW radiation, and more than 20% of this increase is from the clear sky.
Fig. 5 shows the calculated time-developments from 1850 to 2100 of some of the quantities on Table
2, without the adjustment due to the model top being slightly lower than the TOA seen from satellites,
including the three RCP-driven climate projections from 2005 to 2100. Curves for NorESM1-M are
included for reference. NorESM1-Happi experience a higher SW and LW heat fluxes at TOA, but the
net radiative imbalance in the two models is almost identical from 1850 to 2100. Nevertheless, the
global mean 2m-temperature is lower by ~0.3 K in NorESM1-Happi, yet the modelled global
precipitation rate is higher by ca. 0.05 mm d$^{-1}$ (Table 3), which is consistent with the higher latent heat
flux (Table 2).
The budgets in the third column from the right in Table 6 can be compared to a corresponding column
in Table 5 from Iversen et al. (2013). While ca. 10 units (1 unit = 10$^3$ km$^3$ yr$^{-1}$) of water vapour are
evaporated from the oceans in NorESM1-Happi, the model difference in evaporation minus
precipitation (E-P) integrated over the world oceans is considerably smaller. Ca 80% of the added
oceanic evaporation is thus re-cycled back to the oceans. This cools and increases the salinity of the
upper ocean waters where the evaporation increase is large, which may contribute to the thermohaline
forcing and the strength of the AMOC. However, this effect is also influenced by the patterns of the
increased precipitation over ocean. Only 2 more units of water vapour are estimated to be transported
from the oceans to the continents in NorESM1-Happi, which are returned to the ocean as increased
river run-off. These 2 units add to 3 more which are recycled from continental evaporation, and thus
there are 5 more units of precipitation over continents in NorESM1-Happi than in NorESM1-M. In
summary, NorESM1-Happi has a slightly faster cycling of fresh water than NorESM1-M. The positive
biases diagnosed by comparing with GPCP-data in Table 3 indicate, however, that it may be too fast in
both models. The reason for these biases can be associated with the underestimated cloud cover, which
allows more direct sunlight to reach the ground than in reality. The cold bias in the lower atmosphere
still reduces the ability of the model atmosphere to keep the water vapour in the air, thus the efficient
recycling both over oceans and over continents.
Fig. 6 shows the global average 2m temperature relative to the 1851-1980 average for the Hist-
ensemble prolonged to 2015 with NorESM1-Happi, compared to the observationally based global
time-series from NASA-GISS (Hansen et al., 2010). Although the model slightly underestimates the
temperature maximum around 1950, there is good agreement after 1950. The calculated continental
2m temperatures are compared to the observationally based map from Climate Research Unit



(Mitchell and Jones. 2005). Except for in Europe and western parts of Asia, there is a widespread
under-estimate, hence the cold model bias. Fig. 7 compares the model-calculated fluxes of sensible
and latent heat with the FLUXNET Model Tree Ensembles (MTE) estimates (Jung et al., 2011), which
are restricted to vegetated land surface. It is striking that the sensible heat generally is too small, while
the latent heat, and thus the evaporation, is largely over-estimated at low latitude continents,
corroborating the argument about the low cloudiness allowing more solar radiation to reach the ground
and thus cause evaporation. The maxima in overestimations in Africa, America and Australia are
reduced relative to NorESM1-M, but there are increases in southern Europa and western parts of Asia.
The differences for sensible heat are very minor.
Fig. 8 shows evaluations of SST and SSS. The cold bias dominates the pictures also over oceans,
although there are exceptions along the SH storm-belt and in north-west and south-east Atlantic
Ocean. The amplitude of the positive biases is reduced in NorESM1-Happi compared to NorESM1-M
(Fig 12b in Bentsen et al., 2013), while the negative biases are almost the same. Hence the larger
underestimate of the global average (Table 3). The pattern of SSS-biases is almost the same as for
NorESM1-M (Fig 12c in Bentsen et al, 2013). SSS is considerably over-estimated at high latitudes in
the NH, while in the tropics and sub-tropics it is under-estimated.
The modelled and observed sea-ice extent for the recent few decades listed in Table 4, indicate
reduced biases in the simulation results from NorESM1-Happi compared to NorESM1-M, except for
the SH September maximum where the bias are larger. The variability in the data is represented with
the standard deviation, which is generally smaller or equal to the model biases. Both models have
positive biases except during the NH March maximum when the sea-ice cover is under-estimated,
mostly due to too little ice in the Labrador Sea (Fig. 9). The thickness of the sea-ice is shown together
with the extension in Fig. 9. The distribution of the Arctic sea-ice thickness is considerably improved
in NorESM1-Happi, with a pronounced cross-polar gradient with maximum thickness on the
Greenland-Canadian side (approaching 4 m) and a minimum along the Siberian coast. This
improvement was already mentioned in Seland and Debernard (2014).
Table 3 clearly documents that cloud cover is under-estimated in both NorESM1 versions, and that the
negative bias is increased to ca. 20% cloud cover units (the relative error is approximately 33%) with
the 1 degree version of NorESM1-Happi. The zonal averages in Fig. 10 show that the bias exists at all
latitudes, except close to the South Pole where the bias is positive. There are also too much low clouds
and fog in the winter Arctic atmosphere (not shown), which is a persistent problem in many climate
models, including the CCSM4 (Gent et al., 2011). The underestimated cloudiness is to some extent
compensated by exaggerated liquid water contents in the clouds, making them more opaque w.r.t.
radiation, except in the sub-tropics. Notice, however, that the values for liquid water path in Fig. 10b
are valid over oceans only, and the exaggeration is particularly pronounced over the marine





extratropical cyclone regions as discussed by Jiang et al. (2012). The zonally averaged precipitation
rate for winter and summer indicate good agreement with the Legates data (Legates and Willmott,
1990) in the tropics and with the GPCP (Global Precipitation Climatology Project, Adler et al., 2003)
elsewhere. The overestimates shown in Table 3 seem to originate in the tropics.
In summary, by the end of the 20[th] century the near-surface air temperature is simulated too
low by about 1.1-1.2 K (0.8-0.9 K in NorESM1-M) globally and 1.4-1.5 K (1.1-1.2 K) over
land. The global precipitation is estimated to be up to about 0.2 mm day$^{-1}$ (0.15 mm day$^{-1}$) too
high, the evaporation from oceans is over-estimated by ca. 6% (4%), and the net flux between
oceans and continents are ca 12% (8%) over-estimated. The intensity of the water-cycle is
therefore even more overestimated in NorESM1-Happi than in NorESM1-M. Still, the
deduced atmospheric lifetime of water vapour is close to estimates based on Trenberth et al.
(2011), since fraction (E-P)/E over the oceans, which defines the fraction of oceanic
evaporation which is transported and precipitated over continents, is only 0.6% larger than
numbers from Trenberth et al. (2011). It has to be stressed that the model's faster hydrological
cycle is linked to increased lengths and areas of dry spells, due to the coupling between
precipitation and release of latent heat in the atmosphere which produces a positive feedback
with the vertical atmospheric circulation.
In order to attribute climate change and variability since 1850 to possible causes, a standard
set of single forcing simulations are made. In "GHG only", all but the prescribed greenhouse
gas concentrations are kept constant at the 1850-level; in "Aerosol only" all but aerosol
emissions are as in 1850; and in "Natural forcing only", only the natural contributions to the
forcing are varied after 1850. Figure 11 shows results for the global reference height air
temperature and for precipitation. The simulated warming since the 1970s can hardly be
reproduced with natural forcing only, and the greenhouse gases alone would lead to an
exaggerated warming since aerosols significantly dampen the warming. The signals for global
precipitation are less clear, even though they follow those for temperature. As discussed in
Iversen et al. (2013) the regional variations in the simulated precipitation changes are crucial,
since a more intense hydrological cycle in a warmer climate also leads to reduced annual
precipitation and more droughts in some regions while the precipitation intensity increases
(Giorgi et al. 2011). Iversen et al. (2013) discussed this in more detail.





## 4.2 Simulated variability and patterns

There are many patterns of variability of importance for the earth's climate and associated weather. Here we select to summarize the diagnosis of a few which are important for the NH, either directly or through apparent teleconnections. The present analysis is not meant to be complete.

The *Madden–Julian oscillation (MJO)* is the dominant mode of 30–90 days variability in the tropical atmosphere (Madden and Julian, 1971; Zhang, 2005) characterized by large-scale regions of enhanced and suppressed convection that propagate slowly eastward along the equator. The MJO interacts with several large-scale climate phenomena including ENSO (Hendon et al., 2007), and potentially extratropical variability and the North Atlantic Oscillation (NAO) (Cassou, 2008). The CCSM4 was one of the first global climate model that had a fair representation of MJO (Subramanian et al., 2011), and this was also the case in NorESM1-M (Bentsen et al., 2013). Fig. 12 shows the wavenumber–frequency spectra for the NH winter for the equatorial 850 hPa zonal wind U compared to ERA-Interim, and the outgoing longwave radiation (OLR) compared to NOAA satellite data. Although the model has too much energy on shorter periods than a month, the dominant maximum is between 40 and 60 days for wavenumber 1. The spectra are improved compared to NorESM1-M, in particular for OLR.

*Extratropical cyclones* are important vehicles for the atmospheric meridional transport of heat, humidity and momentum between the low and high latitudes, as well as the maintenance of the jet-streams themselves (e.g. Bratseth, 2001; 2003). *Persistent blocking* of eastward propagating cyclones can be important for the occurrence of droughts. The climatological storminess in the NH extratropics is diagnosed by taking the standard deviation of band-pass filtered 500 hPa geopotential height over 2.5 – 6 days (Blackmon, 1976). We refer to the patterns diagnosed in this way as a measure of *extratropical cyclone activity*, even though propagating ridges are included in the variability. Figure 13 shows the evaluation of the seasonal cyclone activity in the Hist-ensembles for 1976-2005 of NorESM1-Happi and NorESM1-M by comparison with the diagnoses of the ERA-Interim data for 1979-2008 (Dee et al., 2011). The cyclone activity is clearly under-estimated in all seasons in both model versions. However, the bias is considerably reduced in NorESM1-Happi, in particular over the Pacific Ocean in all seasons. Reduced bias is also seen where the Atlantic storm-tracks extends into the Nordic sea towards the Arctic, which is important for the weather and precipitation distribution in Europe. The under-estimate over the North Atlantic during winter is, however, considerable also in NorESM1-Happi. We believe most of the improvements in NorESM1-Happi can be ascribed to higher spatial resolution (e.g. Jung et al., 2012), and that the general under-estimated cyclone activity indicates that better resolution is needed in climate models.



*Extratropical blocking* is closely connected with persistent anticyclones, which can suppress
precipitation at mid-latitudes for periods of up to several weeks. The ability of climate models
to simulate and project the climatic occurrence of droughts in mid-latitudes is conditioned by
their ability to simulate blocking. We use the same index as in Iversen et al. (2013) to identify
NH extratropical blocking events. The TM-index (Tibaldi and Molteni, 1990) uses a
persistent reversal of the meridional gradient of the 500 hPa geopotential height around $50^{\circ}$N
as an indicator. Persistence was of at least 5 days was required and the reversed flow had to
be present at 7.5 degrees consecutive longitudes. Pelly and Hoskins (2003) relaxed the
requirement of predefined central blocking latitudes in order to reduce spurious detection. The
vTM index thus allows the central latitude to vary with the longitude of the climatological
storm track, by defining the central latitude where the maximum of the standard deviation of
the 2.5-6 days band-pass filtered geopotential height anomalies at 500 hPa are found. To
account for the seasonal cycle of the cyclone activity, the central latitude for a given month is
calculated as the climatological 3-month moving average centred on the given month.
Fig. 14 shows the seasonal blocking frequency for the Hist-ensemble of NorESM1-Happi and
NorESM1-M simulation for 1976-2005 compared to the ERA-Interim reanalysis for 1979-
2008 (Dee et al., 2011). NorESM1-Happi shows better results than NorESM1-M, although
important systematic errors still persists in several sectors and seasons. Best results are seen in
the autumn (SON), when the occurrence is close to perfect all places except for an under-
estimate in Atlantic Ocean. The errors are also smaller during spring (MAM), while in winter
the systematic under-estimate in the Atlantic-European sector is only slightly reduced. The
results are better over Europe in summer (JJA), but blocking over central Eurasia is under-
estimated and over-estimated in the Pacific sector. As discussed in Iversen et al. (2013), poor
model resolution seems to cause systematic errors of blocking occurrence.
*The El Niño–Southern Oscillation (ENSO)* is a dynamical feature that involves the ocean–atmosphere
interactions with major impacts on the climate variability of the tropical Pacific on seasonal to inter-
annual timescales (Wallace et al., 1998), and with a strong association to global scale patterns and
interaction (Trenberth et al., 1998; Straus and Shukla, 2002). NorESM1-M was amongst 9 out of 20
CMIP5 models that described both modes of the ENSO variability over the central and eastern tropical
Pacific Ocean (Kim and Yu, 2012). As in Bentsen et al., (2013), we use the de-trended monthly SST
anomalies of the NINO3.4 region ($5^{\circ}$S - $5^{\circ}$N; $170^{\circ}$W - $120^{\circ}$W) to identify ENSO variability. The
NINO3.4 index is obtained by normalizing these SST anomalies by their long-term standard deviation.
Fig. 15 shows time series of the NINO3.4 index for the HadISST data set and for Hist1 of NorESM1-
Happi for the years 1900–2005. Also shown are SST anomalies from the corresponding years of the
1850 piControl. The standard deviation of NorESM1-Happi Hist1 and piControl are 0.55 K (0.92 K)
and 0.63 (0.86 K) respectively, where the numbers in brackets are for NorESM1-M. While both
numbers were larger than for HadISST (0.75 K) in NorESM1-M, they are now smaller for NorESM1-
Happi. The frequency of the ENSO variability is reduced in NorESM1-Happi and closer to the
HadISST, as a sharp peak in the frequency spectrum for NorESM1-M for 3 years periods was not
present for NorESM1-Happi (not shown).
### 4.3    Climate sensitivity and Climate Projections
A basic reason for developing global earth system and climate models is for estimating the possible
impact of changes in external forcing on the climate. One single quantity is often referred to in order
to describe to what extent the climate responds to changes that leads to a radiative forcing, the climate
sensitivity. Most measures of climate sensitivity are single numbers that give the response of the
global near-surface air temperature to a standard specified TOA radiative forcing. Here we use
standard protocols from CMIP5 also adopted as in CMIP-DECK for qualifying a model for CMIP6
(Eyring et al., 2016). Two 150-year long runs with NorESM1-Happi start in year 920 of the piControl,
the same start year as Hist1. The "abrupt 4xCO$_2$" (quadrupling of atmospheric CO$_2$ concentrations at
t=0) is used to quantify the equilibrium climate sensitivity (ECS), and the "gradual 4xCO$_2$" (1%
increase per year until quadrupling) is used to quantify the transient climate response (TCR).
To calculate the ECS from first principles requires a full climate model run over several thousand
years (Boer and Yu, 2003). ECS is therefore frequently approximated as the difference, $\Delta T_{eq}$, between
equilibrium near surface air temperatures obtained from two runs over a few decades with a slab ocean
model version which use the same calibration for the control and the experimental run. The first
column in Table 5 gives the ECS measured as $\Delta T_{eq}$ after a doubling of the atmospheric CO$_2$
concentrations (as described in Ch. 3). The number for CCSM4 is from Bitz et al. (2012). The 2
degree version was estimated with a smaller sensitivity (3.14 K) than the 1 degree version, while the 1
degree version NorESM1-Happi is less sensitive than the 2 degree version NorESM1-M.
Table 5 also lists estimates of the ECS based on linear regression between simultaneous values of
near-surface air temperature change ($\Delta T(t)$) and TOA radiation imbalance ($\Delta R(t)$) estimated at time *t*
after the abrupt quadrupling of atmospheric CO$_2$ (Gregory et al., 2004). Assuming negligible
contributions from time-varying feedbacks, the slope of the linear regression line between $\Delta R(t)$ and
$\Delta T(t)$ is the overall feedback parameter $\lambda_{reg} = -d\Delta R/d\Delta T$ (in units of W m$^{-2}$ K$^{-1}$), while the
intercept at $\Delta T=0$ approximates the instantaneous forcing $R_{f,reg}$, and the intercept $\Delta T_{reg}$ at $\Delta R = 0$
approximates the ECS. This estimate of $R_f$ disregards rapid adjustment during the first year of the
simulation (Andrews et al., 2012). Table 5 lists all these quantities for the three models valid for CO$_2$



doubling. The independent estimate of the inferred TOA radiative forcing is $R_{f0}$ = 3.5 W m⁻² (Kay et
al., 2012), which is larger than the regression estimates for all three models, although the difference
for NorESM1-Happi is much smaller. The difference is normally ascribed to fast feedbacks during the
first year after the $CO_2$-increase, while the difference between $\Delta T_{reg}$ and $\Delta T_{eq}$ considered to reflect the
error of using linear regression due to slow feedbacks e.g. in the oceans (Senior and Mitchell, 2000).
Murphy (1995) proposed to use the remaining TOA radiative imbalance $\Delta R(t)$ at the time $t$ to
approximate ECS. This approximation, termed the effective climate sensitivity, is:

$$\Delta T_{eff}(t) = \Delta T(t)R_f/\left(R_f - \Delta R(t)\right) \qquad (2)$$

Assuming the same linear relationship between $\Delta T(t)$ and $\Delta R(t)$, $\Delta T_{eff}$ should not depend on time,
but also in this case, slow feedback processes may do. In parallel to $\Delta T_{eff}(t)$ it is possible to define an
effective feedback parameter (Gettelman, 2012) $\lambda_{eff} = R_{f0}/\Delta T_{eff}$, which appears to be more
internally consistent than $\lambda_{reg}$, see Table 5, and we consider -1.22 W m⁻² K⁻¹ to be the more reliable
estimate of the feedback parameter for NorESM1. The values of $\Delta T_{eff}$ in Table 5 are averages of
annual values for the last 40 of the 150 simulation years of the abrupt 4x$CO_2$ experiments, and divided
by 2 for doubling rather than quadrupling. For the three models, $\Delta T_{reg}$ and $\Delta T_{eff}$ are almost equal and
the models are amongst the least sensitive of the CMIP5 models (e.g. Andrews et al., 2012).
The TCR can be estimated by similar simple methods as used for ECS, using the results from the
gradual 4x$CO_2$ experiment.  The globally averaged change in near-surface air temperature at the time
of doubled atmospheric $CO_2$ (averaged over simulation years 60-80). An effective response that takes
into account the remaining TOA radiative imbalance can also be estimated by applying Eq. (2). The
values given in the rightmost columns in Table 5 present these estimates. While the approximate
values for ECS were close, the TRCs differ, with NorESM1-M having the smallest values. The
reasons for this can be related to the strong AMOC (Iversen et al., 2013) and features associated with
spatial resolution, but require further investigations beyond the scope of the present paper.
The experiments designed for estimating ECS and TCR are not meant to reflect realistic scenarios of
future developments. For this purpose, the RCPs were designed for CMIP5, and the fully coupled
NorESM1-Happi have extended Hist1 to year 2100 by assuming the changes defined in RCP 2.6, 4.5
and 8.5, were the numbers are pre-calculated values of the TOA radiative forcing at 2100 relative to
1850. In section 5 we address the targeting of temperature ceilings with simplified methods.
Fig. 5 includes the calculated projections of the fluxes at the top of the model atmosphere, the global
temperature at reference height, and the global precipitation rate. The results indicate that it takes time
until 2040-50 before considerable differences between the scenario projections are seen. This is also
the case for changes in the strength of the Atlantic Meridional Overturning Circulation (AMOC), see



Fig. 16. Notice that the AMOC in NorESM1-Happi towards 2100 for RCP8.5 is still stronger than
many other models for the present climate.
In Fig. 17 we see evidence of Arctic Amplification of the global signals of systematic temperature
change as well as variation. The apparent Arctic variability is likely to be exaggerated, since the
average is taken over less than 5% of the earth's surface (north of 65°N), and the average over any 5%
of the surface has larger variability than the global average. For the climate change signal, a rule of
thumb is that the Arctic temperature change is ca. twice that of the global.
The numbers in Table 6 are valid for the 30 years towards the end of the 21$^{st}$ century compared to the
end of the 20$^{th}$ (approximately). Unfortunately, we have not been able to run several ensemble
members, and we have no reliable estimate of signal-to-noise ratio. A simple "back-of-the-envelope"
calculation shows that RCP2.6 produces a 1.46 K global temperature increase towards 2100 relative to
1850, although this value does not represent a stable new equilibrium as assumed by the Paris
agreement. The average response for the CMIP5 archive is 1.55 K.
For simplicity, taking RCP2.6 with NorESM1-Happi to approximate the situation in a 1.5°C world, the
annual sea-ice area (both hemispheres) will be reduced by 8.2% (12.6%) compared to end 20$^{th}$ century,
the global precipitation amounts will be 2.2% (3.1%) larger, the continental precipitation will be 2.4%
(4.2%) larger, and the continental (P-E) will be 2.7% (6.8%) larger, all compared to end 20$^{th}$ century,
and completely based on the numbers from NorESM1-Happi. In comparison, the RCP4.5 projection
with NorESM1-Happi gives a 2.14 K warmer climate by the end of 21$^{st}$ century than in 1850 (2.40 K
for the CMIP5 archive). Using a similarly simple calculation assuming a factor 0.93 of the changes in
RCP4.5 to approximate a 2.0°C warmer world than in 1850, we arrive at the numbers given in the
brackets above. These numbers are based on single projections from one single model, and
accordingly must be used very carefully. Yet, they indicate that there can be considerable differences
between the climate of a 2.0°C world and a 1.5°C world. We continue to discuss this with more
robustness in the next section. If the RCP8.5 should turn out to be realized, we are facing a 3.7°C
world (according to the single NorESM1-Happi realization), with considerable larger impacts on sea-
ice and precipitation.
**5   Global Temperature Ceilings and Arctic Amplification**
**5.1   NorESM1-HappiAMIP and NorESM1-HappiSO**
With NorESM1-Happi, we have contributed to the experiments in the multi-model HAPPI project.
The target of the experiments is to investigate how important aspects of global warming may manifest
itself under two stabilisation scenarios of 1.5°C and 2.0°C warmer than the 1850 climate. The



challenge is to determine forcing conditions that will produce the temperature targets, and thus also
calculate other variables that characterize the climate state. Given that the climate system takes
centuries and longer to approach new equilibria after sustained changes in the TOA radiation balance,
it is not practically straightforward to apply coupled climate models for this purpose.
Simplified methods are required, but they should be based on the state of knowledge that present
climate and earth system models represent. Sanderson et al. (2017) proposed to emulate the results
from an advanced climate model with simplified methods. In HAPPI, Mitchell et al. (2017) proposed
to prescribe SST, sea-ice, and anthropogenic forcing elements and run models that couple atmosphere
and land but avoid the calculation of slow ocean processes and sea-ice. These are AMIP-type
experiments, and here we will identify our runs with for HAPPI as NorESM1-HappiAMIP.
The construction of driving input data to the AMIP experiments in HAPPI is thoroughly described by
Mitchell et al. (2017). They are observationally based (Taylor et al., 2012) for the present day climate
(PD, 2005-2016), but for the future temperature targets they are achieved by combining forcing data
for RCP2.6 and RCP4.5 for the year 2095, and associated data for SST from the model projections in
the CMIP5 archive. The future sea-ice is not directly taken from the CMIP5-models, partly because of
the huge variation in biases and partly because the models have common biases in the SH. In short, the
sea-ice in any point was fitted to the SSTs after establishing an observationally based linear relation
(by regression) between SST- and sea-ice anomalies for 1996-2015. The NorESM1-HappiAMIP data
cover 100 ensemble members of length 10 years (each after a 1-year spin-up) for PD, for the $1.5^{\circ}$C
world, and for the $2.0^{\circ}$C world, in all 3000 years of data. 25 additional ensemble members per period
have been run with high temporal resolution to enable dynamic downscaling of the results. The data
are available for download at http://portal.nersc.gov/c20c/data.html.
One shortcoming of pure AMIP-type runs where both SST and sea-ice are prescribed, is the missing
possibility to explicitly calculate the effects of feedback mechanisms associated with them. Arctic
amplification (AA) such as seen in evidence for in Fig. 17 is likely to be associated with such
feedbacks. Pithan and Mauritsen (2014) pointed out the lapse-rate and albedo feedbacks as important
causes of AA, and both are associated with low-level temperature changes in the Arctic. We have
therefore designed the slab-ocean model version NorESM1-HappiSO and calibrated the upper ocean
flux divergence of heat ($Q_f$) to emulate the NorESM1-HappiAMIP experiments as described in Ch. 3.
Three 310-year long experiments have been made for, respectively PD, $1.5^{\circ}$C, and $2.0^{\circ}$C. After a spin-
up of 45 years, a new equilibrium is considered reached in each case, leaving three equilibrated runs of
length 265 years. The results of these are compared to the NorESM1-HappiAMIP ensembles.



## 5.2 Arctic Amplification and the 0.5$^o$C difference

In Table 7 we have defined a *Polar Amplification Factor" PAF* $= \Delta T_{Polar}/\Delta T_{Global}$, where the subscript Polar identifies an average over the area poleward of the 60$^o$ latitude circles for each hemisphere, and where T in this case is the 2 m temperature. The table shows results for PAF in both hemispheres together with the global change in 2 m temperature. Results for AMIP and SO are shown together with the single projections for the RCP2.6 and RCP4.5 scenarios. It is reassuring that both NorESM1-Happi AMIP and SO quite accurately hit the temperature targets, although the SO produces a slightly lower PD temperature. The fully coupled NorESM1-Happi confirms the negative temperature bias, and the increments are also somewhat larger. The PAF is considerably larger in the Arctic than in the Antarctic, and the AA is enhanced with the SO model compared to the AMIP. The latter includes the difference between the 2.0 and 1.5$^o$C worlds for which NH_PAF is 18% larger.

Table 8 shows similar statistics as Table 7, but for the NH extratropical winter and summer land temperatures, land precipitation rates, and sea-ice area. The SO model has a tendency to produce a colder winter climate than AMIP, which also manifests itself with a larger sea-ice area. In summer, however, these differences are much smaller. Probably the tendency towards colder climate, which is established as a systematic error for the fully coupled model, shows up during winter with the SO model. Furthermore, the 1.5$^o$C-PD response is a smaller reduction in sea-ice area with SO than in the prescribed data for AMIP for both seasons. For the 2.0$^o$C-PD response this is only seen during winter, while the sea-ice area is more reduced in summer. These features are also reflected in the temperatures and precipitation over land.

The increases in 2 m temperatures and the precipitation over NH land is larger for SO than for AMIP in summer (JJA), but not in winter (DJF). For the difference 2.0$^o$C -1.5$^o$C, the SO response is larger than AMIP for summer precipitation and summer sea-ice area. Otherwise, the differences are negligible, and even of opposite sign for the winter sea-ice area.

The sea-ice area for the PD AMIP-runs in DJF (Table 8) is considerably smaller than the observed for March in Table 4, although also the first is based on observations. The sea-ice area is at its maximum in March, and a difference is to be expected. The difference is considerably smaller for the SO and fully coupled model (less than 50%). Table 4 shows that NorESM1-Happi has a negative bias in March, while Table 8 indicates a positive bias averaged for the three winter months (DJF). Assuming that the observations in the two tables are internally consistent (which is a relevant comment since the time periods for observations are not the same) it appears that Arctic sea-ice cover grows too fast during the early winter months and too slow towards the March maximum. More in-depth studies are required to establish if the early winter over-estimate is due to a cold-bias tendency of NorESM1-HappiSO (as well as in NorESM1-Happi) that is




amplified by the prognostic sea-ice cover in the SO model, or if it reflects a natural process of reduced
feedbacks associated with sea-ice and SST during winter darkness which is not well represented in the
model.

### 5.3    Cyclone activity and blocking

As for the fully coupled NorESM1-Happi, we have calculated the extratropical cyclone activity and
blocking occurrence. Fig. 18 shows the seasonal cyclone activity in the PD AMIP runs compared to
ERA-Interim and the difference in cyclone activity between the SO and AMIP for PD. When
comparing with the corresponding Fig. 13 for the fully coupled NorESM1 models, it should be noted
that the colour scale is finer in Fig 18 by approximately a factor 2 for the plots on the left, and a factor
3 for the plots on the right. The absolute values of the biases are smaller for both the AMIP and SO,
and while it was clear that all biases were negative for the coupled models, i.e. generally too low
cyclone activity, the AMIP and SO results have biases of both signs. A systematic displacement of the
cyclone activity closer to the Arctic than in ERA-Interim is a general impression. In the Atlantic-
European sector, there is too much cyclone activity over the continent in winter and spring, and too
little activity in the branch over the Nordic Seas between Scandinavia and Greenland. These biases are
not reduced in the SO runs. Instead they rather seem to be slightly larger, with a few exceptions. In the
autumn the positive biases in AMIP results over the Arctic are reduced in the SO runs.
Fig. 19 shows the changes in cyclone activity from PD to a 1.5$^o$C world. For reference, results are also
shown for the RCP2.6 of the fully coupled NorESM1-Happi, since RCP2.6 produces approximately
1.5$^o$C higher 2 m global temperatures than the piControl (see Table 6). Except for the summer months
when the response is small, the SO response is considerably larger than the AMIP response, and this is
particularly evident for winter and spring. The signal is dominated by reduced extratropical cyclone
activity, which is also seen in the RCP2.6 scenario of NorESM1-Happi. Except for the autumn
months, the AMIP response appears artificially small, although we do not know how the real response
will be. To the extent that reduced activity can be interpreted as slower propagation of cyclone waves
in the westerlies, this can be associated with reduced low-level meridional temperature gradients
associated with the high-latitude warming over the Arctic (e.g. Francis and Vavrus, 2012; Screen and
Simmonds, 2013). In winter the SO-response include an apparent northward displacement of the
cyclone activity in the Atlantic sector of the Arctic. We have also looked at the difference in cyclone
activity between a 1.5$^o$C and a 2.0$^o$C world for SO and AMIP. The differences are small in all seasons
although there are patches of statistically significant differences (not shown).
Fig. 20 shows seasonal statistics for blocking occurrence (the vTM index) for the AMIP runs at PD,
compared to ERA-Interim results. The results for the SO runs are almost identical. The results are
slightly better than for the coupled NorESM1 models (Fig. 14), but there still are problems in




representing sufficient blocking occurrence in the Atlantic sector. Fig. 21 shows the changed occurrence of blocking for the 1.5°C -PD difference and for the 2.0°C -1.5°C difference. Both AMIP results and SO results are shown. Statistically significant changes (at the 5 % level) are more frequent with the SO runs than the AMIP, and the signals of changed blocking occurrence are also considerably stronger in the SO results (hence the different units on the y-axes). However, there are several cases for which the AMIP runs show increased (decreased) blocking occurrence with statistical significance, which are not confirmed by increased (decreased) blocking occurrence in the results from the SO model. For example, in winter for the 1.5°C -PD difference, the AMIP-results predict a statistically significant reduction in blocking occurrence at 240°E, while the SO-results predict a strong and statistically significant increase in blocking occurrence. The level of significance is 5%, so there still is a possibility that this is by chance, but it can be a consequence of the feedbacks with SST and sea-ice allowed in the SO-model.

Considering the difference between a 1.5 and a 2.0°C world, the SO results predict reduced occurrence of blocking in Europe in winter and autumn and increased occurrence in central Eurasia and the Pacific Ocean. The strongest signal is seen in Atlantic-European sector in spring, where a statistically significant increase from PD to the 1.5°C world is strengthened by a further statistically significant increase from 1.5 to 2.0°C. There are several cases with considerable changes of blocking occurrence between a 1.5 and a 2.0°C world, e.g., a reduced occurrence in autumn between 20 and 60°E, and an increased occurrence around 90°E in summer.

### 5.4 Arctic sea-ice reduction

The extension, thickness and concentration of sea-ice are important variables in the climate system. Fig. 22 show the extension and concentration of Arctic sea-ice in March and September calculated with NorESM1-HappiSO. For PD the concentration map from the model is compared to observational data from OSI-SAF (2017). The quality of the model-data is better in March than in September, when the model seems to underestimate the ice-concentration. For the temperature targets, the sea-ice in March is predominantly reduced along the edges. However, it should be borne in mind that the SO model may be too cold (Table 8). In September, the reduction in sea-ice concentration (and extension) is moderate for the 1.5-PD change, but the further reduction from 1.5 to 2.0°C is substantial.

The latter is indeed confirmed by Fig. 23 which shows a histogram of SO-model calculated relative occurrence of NH September sea-ice in different classes of extent for PD and the two temperature targets. For PD the model tends to produce too few cases with largest extent. Looking at the possibility of having ice-free Arctic September minima, i.e. the fraction of cases in the class between 0 and $1 \times 10^6$ km$^2$, the model predict probability of this occurrence to be practically zero under PD conditions, a tiny fraction in the 1.5°C world, and ca. 40 % for the 2.0°C world. The difference between the two





temperature targets is therefore potentially very large for the Arctic sea-ice in summer-fall, a result
that was also found by Sanderson et al. (2017).
**6    Summary and Conclusions**
This paper presents a validation of model tools which are used to study the significance of differences
in the global climate between a world with a stable 2 m temperature which is 1.5$^o$C higher and 2.0$^o$C
higher than in 1850. The incitement is the outcome of the Paris protocol that a Special Report from
IPCC is under development. The state-of-science model tools we have used for this purpose is
basically a slightly upgraded versions of the atmosphere and land models in the coupled NorESM1-M,
which was used for CMIP5 (Bentsen et al., 2013; Iversen et al., 2013, Kirkevåg et al., 2013).
A proper validation of such tools requires a documentation of its performance fully coupled to a deep
ocean model and a sea-ice model. The first part of results for this paper (Ch. 4) presents a selection of
results from a quite complete set of the fully coupled model runs with NorESM1-Happi. The name
Happi comes from our association with the multi-model HAPPI project (Mitchell et al., 2016 and
2017), for which large ensembles of AMIP-type experiments are made and analysed. The second part
of results (Ch. 5) focuses results from the HAPPI AMIP runs with the atmosphere and land modules of
NorESM1-Happi, complemented with results from a specially designed slab-ocean (SO) version of the
model (NorESM1-HappiSO) which includes a fully dynamic-thermodynamic sea-ice module. Our
main reason for complementing with a SO model, is that the AMIP-runs pre-define aspects of the
Arctic Amplification (AA) which do not allow adjustments by on-line feedbacks. The SO-model
allows changes in SST and sea-ice which can influence both surface albedo and the atmospheric
temperature lapse rate, which both are major elements in producing AA (Pithan and Mauritsen, 2014).
The results and discussions in Ch. 4 show that NorESM1-Happi is a valid model tool for climate
research at the present state of science. The stability of long-term climate properties over the 500-year
pre-industrial run is at least as good as for NorESM1-M, except for the AMOC which is strong and
has a positive trend. The model has double resolution and a few improvements in physical
parameterizations compared to the original CMIP5 version. This partly increases some of the biases,
cloud cover, 2 m temperature, and the strength of the AMOC in particular, but considerably improves
the Arctic sea-ice thickness and reduces the biases in extratropical cyclone activity and occurrence of
blocking in the northern hemisphere. Although the intensity of the water cycle is exaggerated (by
6.7% globally), the atmospheric residence time of water vapour and the budget of transport between
oceans and continents are reasonable. The model represents the Madden-Julian Oscillation with
reduced errors than in NorESM1-M, and the return period of ENSO events is slightly reduced, but
with too low amplitudes. The model's standard estimates of climate sensitivity are still in the lower

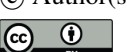



range amongst global climate models. With reference to the question of differences between a 1.5$^{\circ}$C
and a 2.0$^{\circ}$C world, NorESM1-Happi shows a considerable difference in response between the RCP2.6
and the RCP4.5 scenarios.
In Ch. 5 we saw that the NH extratropical cyclone activity is not improved in the SO model results
compared to the AMIP results, and for occurrence of NH blocking the differences are negligible. The
SO results also show stronger responses than the AMIP results to a global change. In particular, the
SO result reduces the NH extratropical cyclone activity considerably. Changes in the occurrence of
blocking are larger, although the total signals of change in response in a warmer world are mixed. The
SO-model simulates considerable differences in the reduction of sea-ice in the Arctic between a 1.5$^{\circ}$C
and a 2.0$^{\circ}$C world. Ice-free summer conditions in the Arctic is estimated to be very rare for the 1.5$^{\circ}$C
case, but to occur 40% of the time for the 2.0$^{\circ}$C case. These results agree with some fully coupled
models, but need to be further confirmed.
**7   Code and Data Availability**
The source code for NorESM1-Happi is not open for anyone to download, since parts of the code is
imported from several code development centres. The code can be made available within the
framework of an agreement. Data from the model experiments made with the model can be made
available as well, e.g. NCC / NorESM1-HAPPI from http://portal.nersc.gov/c20c/data.html. Contacts:
oyvindse@met.no and ingo.bethke@uni.no.
**Acknowledgements**
This work received financial support from the Norwegian Research Council, project no. 261821
(HappiEVA), and in-kind support from Geophysical Institute, Univ. of Bergen, UniResearch, Bergen,
and from Met Norway. Jan Fuglestvedt, Cicero, Norway, Dann Mitchell, Univ. Bristol, UK, and
Myles Allen, Univ. Oxford, UK, were instrumental in inspiring the initiating this work, and have
strongly contributed in discussions. HPC-resources for the NorESM model runs were provided in kind
from Bjerknes Centre for Climate Research and MET Norway. Storage for NorESM-data was
provided through Norstore/NIRD (ns9082k). The development of NorESM has been possible because
of the granted early access to the later public versions of the CCSM4 and CESM1 by the US National
Center for Atmospheric Research (NCAR).



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

28   **Tables**

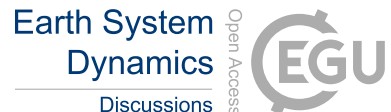

**Table 1**. Calculated global budgets for the on-line modelled aerosol components and precursors in an
AMIP experiment with NorESM1-Happi averaged over the 10 years 2006-2015 (Happi). The budget
is compared to an on-line run with NorESM1-M over the same years with RCP8.5-emissions for 2006-
2015 (M), and an off-line aerosol experiment with NorESM1-M valid for year 2000 (EmPD2000 in
Kirkevåg et al., 2013) (M$_{2000}$).

| Comp. | NorESM1 version | Emis. Tg/a | Source Tg/a | Fraction of sink | | Fraction of chem. loss 1: as MSA 2: in clear air % | Global burden Tg | Atmosph. residence time d |
|---|---|---|---|---|---|---|---|---|
| | | | | as wet dep % | as chem. loss % | | | |
| DMS | Happi | 18.0 | 18.0 | 0 | 100 | 27.0 (1) | 0.12 | 2.43 |
| | M | 18.1 | 18.1 | 0 | 100 | 27.5(1) | 0.12 | 2.38 |
| | M$_{2000}$ | 18.0 | 18.0 | 0 | 100 | 27.1(1) | 0.12 | 2.40 |
| SO$_2$-S | Happi | 64.7 | 78.1 | 8.4 | 67.7 | 16.1 (2) | 0.25 | 1.16 |
| | M | 63.2 | 76.4 | 8.1 | 69.8 | 15.9 (2) | 0.23 | 1.11 |
| | M$_{2000}$ | 66.3 | 79.5 | 7.9 | 69.1 | 15.4 (2) | 0.24 | 1.11 |
| SO$_4$-S | Happi | 1,66 | 54.6 | 92.3 | 0 | | 0.55 | 3.68 |
| | M | 1.62 | 55.1 | 92.1 | 0 | | 0.55 | 3.61 |
| | M$_{2000}$ | 1.70 | 56.8 | 91.6 | 0 | | 0.59 | 3.80 |
| BC | Happi | 8.58 | 8.58 | 78.1 | 0 | | 0.14 | 5.73 |
| | M | 7.65 | 7.65 | 72.8 | 0 | | 0.16 | 7.61 |
| | M$_{2000}$ | 7.70 | 7.70 | 71.9 | 0 | | 0.17 | 8.12 |
| POM | Happi | 123.6 | 138.2 | 80.4 | 0 | | 2.38 | 6.24 |
| | M | 120.1 | 135.0 | 79.2 | 0 | | 2.55 | 6.83 |
| | M$_{2000}$ | 122.2 | 137.0 | 78.6 | 0 | | 2.87 | 7.58 |
| Sea-Salt | Happi | 7397 | 7397 | 46.3 | 0 | | 5.21 | 0.26 |
| | M | 6716 | 6716 | 47.7 | 0 | | 4.81 | 0.26 |
| | M$_{2000}$ | 6462 | 6462 | 45.4 | 0 | | 4.94 | 0.28 |
| Dust | Happi | 1666 | 1666 | 24.6 | 0 | | 12.10 | 2.64 |
| | M | 1674 | 1674 | 23.7 | 0 | | 12.11 | 2.63 |
| | M$_{2000}$ | 1672 | 1672 | 25.2 | 0 | | 11.73 | 2.55 |

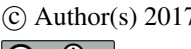



**Table 2.** Global and annual averages calculated for the two versions of the model (for 1976 - 2005 of
Hist1 for NorESM1-Happi and NorESM1-M) and for observationally based or reanalyzed data
(references below). The NorESM values are adjusted since the top of the model is slightly below the
TOA seen from satellites (Collins et al., 2006). The actual fluxes at the top of the model is shown in
Fig. 5, and the net radiative flux at the top of model for Hist1 (1976 – 2005) is about +0.61W $m^{-2}$ in
NorESM1-Happi and +0.54W $m^{-2}$ in NorESM1-M.

| Variable (unit) | NorESM1-Happi 1 deg. resolution | NorESM1-M 2 deg. resolution | OBS / reanalysis |
|---|---|---|---|
| TOA net SW flux (W $m^{-2}$) | 240.2 | 234.9 | 240.6[a] 244.7[b] 234.0[c] |
| TOA net clear-sky SW flux (W $m^{-2}$) | 289.4 | 289.5 | 287.6[a] 294.7[b] 289.3[c] |
| TOA upward LW flux (W $m^{-2}$) | 237.6 | 232.4 | 239.6[a] 239.0[b] 233.9[c] |
| TOA clear-sky upward LW flux (W $m^{-2}$) | 263.5 | 262.3 | 266.1[a] 266.9[b] 264.4[c] |
| TOA LW cloud forcing (W $m^{-2}$) | 25.81 | 29.90 | 26.48[a] 27.19[b] 30.36[c] |
| TOA SW cloud forcing (W $m^{-2}$) | -49.20 | −54.57 | −47.07[a] −48.59[b] −54.16[c] |
| Cloud cover (%) | 46.36 | 53.76 | 66.80[d] 66.82[e] |
| Cloud liquid water path (g $m^{-2}$) | 121.3 | 125.3 | 112.6[f] |
| Surface sensible heat flux (W $m^{-2}$) | 18.0 | 17.8 | 19.4[h] 15.8[i] 13.2[j] |
| Surface latent heat flux (W $m^{-2}$) | 83.7 | 81.7 | 87.9[h] 84.9[k] 82.4[g] 89.1[l] |

[a]CERES-EBAF (Loeb et al., 2005, 2009, 2012); [b]CERES (Loeb et al., 2005, 2009,2012), [c]ERBE (Harrison et al., 1990; Kiehl and Trenberth, 1997), [d]ISCCP (Rossow and Schiffer, 1999; Rossow and Dueñas, 2004), [e]CLOUDSAT (L'Ecuyer et al., 2008), [f]MODIS (Greenwald, 2009; Seethala and Horváth, 2010), [g]ERA40 (Uppala et al., 2005), [h]JRA25 (Onogi et al., 2007), [i]NCEP (Kanamitsu et al., 2002), [j]LARYA (Large and Yeager, 2004, 2008), [k]ECMWF (Trenberth et al., 2011), [l]WHOI (Yu and Weller, 2007; Yu et al., 2008).



**Table 3**. Bias and RMS error for selected atmospheric variables in the Hist1 (1976-2005) from
NorESM1-M and NorESM1-Happi. Obs. Data: Cloudsat (L'Ecuyer et al., 2008); CRU (Mitchell and
Jones, 2005; Morice et al., 2012); HadISST (Hurrel et al., 2008); GPCP, (Adler et al., 2003; Huffman
et al., 2009); ISCCP (Rossow and Schiffer, 1999; Rossow and Dueñas, 2004). Numbers are produced
by the diagnostic package from NCAR.

| | | Total cloud cover    % | | $T_{2m}$ continents CRU            K | SST HadISST K | Total Precip. GPCP            mm $d^{-1}$ |
|---|---|---|---|---|---|---|
| | | Cloudsat | ISCCP | | | |
| Happi | bias | -20.46 | -20.44 | -1.45 | -0.56 | 0.19 |
| | rmse | 22.08 | 23.01 | 2.40 | 1.19 | 1.15 |
| M | bias | -13.06 | -13.04 | -0.88 | -0.25 | 0.13 |
| | rmse | 15.64 | 17.40 | 2.38 | 1.24 | 1.22 |





**Table 4**. NH and SH sea-ice extent and standard deviation (both in $10^6$ km$^2$) calculated for March and
September in the Hist1 runs with NorESM1-M and NorESM1-Happi and from observations (OSI-
SAF, 2017) for the years 1979–2005. The best estimate ice concentration fields from the observations
were interpolated to the model grid before the extents were calculated.

| | | | Extent ($10^6$ km$^2$) | Std ($10^6$ km$^2$) | Deviation ($10^6$ km$^2$) |
|---|---|---|---|---|---|
| NH | March | OBS | 15.41 | 0.42 | 0 |
| | | M | 14.34 | 0.25 | -1.09 |
| | | Happi | 14.72 | 0.26 | -0.69 |
| | Sept. | OBS | 6.9 | 0.62 | 0 |
| | | M | 8.49 | 0.4 | 1.59 |
| | | Happi | 7.95 | 0.40 | 1.05 |
| SH | March | OBS | 4.46 | 0.41 | 0 |
| | | M | 5.65 | 0.46 | 1.19 |
| | | Happi | 5.17 | 0.51 | 0.71 |
| | Sept. | OBS | 18.76 | 0.34 | 0 |
| | | M | 19.25 | 0.50 | 0.49 |
| | | Happi | 19.98 | 0.44 | 1.22 |

**Table 5**. Three estimates of equilibrium climate sensitivity (ECS) for abrupt doubling of atmospheric
CO$_2$ concentrations, two estimates of the overall feedback parameter, and two estimates of the
transient climate response (TCR) of NorESM1-Happi with 1$^o$ resolution, compared to NorESM1-M
with 2$^o$ resolution (Iversen et al, 2013), and for CCSM4 with 1$^o$ resolution (Bitz et al., 2012).

| Model | $\Delta T_{eq}$ K | $\Delta T_{eff}$ K | $\Delta T_{reg}$ K | $R_{f\_reg}$ W m$^{-2}$ | $\lambda_{reg}$ W m$^{-2}$K$^{-1}$ | $\lambda_{eff}$ W m$^{-2}$K$^{-1}$ | $\Delta T_{TCR}$ K | $\Delta T_{TCR,eff}$ K |
|---|---|---|---|---|---|---|---|---|
| CCSM4, 1 deg. | 3.20 | 2.78 | 2.80 | 2.95 | -1.053 | -1.260 | 1.72 | 2.64 |
| NorESM1-M 2 deg | 3.50 | 2.86 | 2.86 | 3.17 | -1.108 | -1.224 | 1.37 | 2.29 |
| NorESM1-Happi 1 deg | 3.34 | 2,87 | 2.82 | 3.43 | -1.214 | -1.220 | 1.52 | 2.47 |


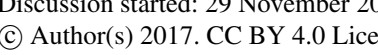


**Table 6**. Simulated changes in selected global annual data with NorESM1-Happi from the period
1976-2005 (Hist1) to 2071-2100 based on three projected representative concentration pathways
(RCP) scenarios. Column no. 3 from the right are total values simulated for 1976-2005 (Hist1) where
the water-cycle values can be compared to the budget values from Trenberth et al. (2011) in the
rightmost column. Column no. 2 from the right gives the differences between Hist1 (1976-2005) and
the 1850 piControl.

| | RCP8.5 – Hist1 | RCP4.5 – Hist1 | RCP2.6 – Hist1 | Hist1 1976-2005 | Hist1 – piControl | Water Budget 2001-2008 |
|---|---|---|---|---|---|---|
| $T_{2m}$ (K) | +3.09 | +1.59 | +0.91 | 286.44 | +0.55 | - |
| SST (K) | +2.02 | +1.03 | +0.59 | 290.88 | +0.36 | - |
| $AREA_{SeaIce}(10^6 \, km^2)$ | -6.6 | -2.8 | -1.7 | 20.7 | -1.0 | - |
| $P_{GLOBAL}$ 1000 $km^3 \, yr^{-1}$ | +28.1 | +17.5 | +11.9 | 533.5 | +1.5 | 500 |
| $E_{OCEANS}$ 1000 $km^3 \, yr^{-1}$ | +26.1 | +15.1 | +10.1 | 451.7 | +1.3 | 426 |
| $(E-P)_{OCEANS}$ 1000 $km^3 \, yr^{-1}$ | +7.5 | +3.3 | +1.2 | 45.2 | +0.6 | 40 |
| $P_{OCEANS}$ 1000 $km^3 \, yr^{-1}$ | +18.2 | +11.8 | +8.9 | 406.5 | +0.7 | 386 |
| $P_{LAND}$ 1000 $km^3 \, yr^{-1}$ | +9.9 | +5.7 | +3.0 | 127.0 | +0.8 | 114 |
| $E_{LAND}$ 1000 $km^3 yr^{-1}$ | +2.5 | +2.4 | +1.8 | 81.8 | +0.2 | 74 |



**Table 7**. NH and SH Polar Amplification Factor (PAF= $\Delta T_{Polar}/\Delta T_{Global}$, polar is the poleward side
of the 60° latitude) and the global temperature at reference height (2m) are listed for PD (2006-2015)
together with their increments for a 1.5 °C and a 2.0 °C warmer world than the 1850 pre-industrial.
Results are taken from the NorESM1-Happi AMIP-type runs and the NorESM1-HappiSO. Similar
results from the RCP2.6 and RCP4.5 projections at 2071-2100 with NorESM1-Happi compared to
Hist1 (1976-2005) are included for reference.

|  | Period or Difference | NH_PAF | SH_PAF | $T_{2m}$ K |
|---|---|---|---|---|
| NorESM1-Happi AMIP 100x10 years | PD |  |  | 287.30 |
|  | 1.5 - PD | 2.34 | 1.62 | +0.71 |
|  | 2.0 - PD | 2.17 | 1.35 | +1.20 |
|  | 2.0 – 1.5 | 1.93 | 0.95 | +0.49 |
| NorESM1-Happi Slab Ocean 265 years | PD |  |  | 287.17 |
|  | 1.5 - PD | 2.49 | 1.79 | +0.68 |
|  | 2.0 - PD | 2.41 | 1.60 | +1.19 |
|  | 2.0 – 1.5 | 2.27 | 1.30 | +0.51 |
| NorESM1-Happi Coupled 30 years | Hist1 |  |  | 286.44 |
|  | RCP2.6 – Hist1 | 3.00 | 0.29 | +0.91 |
|  | RCP4.5 – Hist1 | 2.70 | 0.58 | +1.59 |
|  | RCP4.5 – RCP2.6 | 2.29 | 0.97 | +0.68 |



1    **Table 8**. Similar as Table 7, but for 2m temperature over land, precipitation on land, and sea-ice area

2    in the northern, extratropical hemisphere (20-90 °N) in winter (DJF) and summer (JJA).

|  | Period or Difference | $T_{Land}^{DJF}$ K | $T_{Land}^{JJA}$ K | $P_{Land}^{DJF}$ mm d$^{-1}$ | $P_{Land}^{JJA}$ mm d$^{-1}$ | $AREA_{SeaIce}^{DJF}$ $10^6$ km$^2$ | $AREA_{SeaIce}^{JJA}$ $10^6$ km$^2$ |
|---|---|---|---|---|---|---|---|
| NorESM1-Happi AMIP 100x10 years | PD | 264.89 | 293.60 | 1.212 | 2.538 | 11.90 | 6.31 |
|  | 1.5 - PD | +1.43 | +0.81 | +0.067 | +0.09 | -0.99 | -0.56 |
|  | 2.0 - PD | +2.33 | +1.66 | +0.093 | +0.13 | -1.39 | -0.89 |
|  | 2.0 – 1.5 | +0.90 | +0.84 | +0.026 | +0.04 | -0.40 | -0.33 |
| NorESM1-Happi Slab Ocean 265 years | PD | 264.13 | 293.42 | 1.214 | 2.549 | 13.33 | 6.26 |
|  | 1.5 - PD | +1.40 | +1.24 | +0.041 | +0.054 | -0.51 | -0.34 |
|  | 2.0 - PD | +2.24 | +2.00 | +0.063 | +0.073 | -0.75 | -1.18 |
|  | 2.0 – 1.5 | +0.85 | +0.77 | +0.022 | +0.019 | -0.24 | -0.84 |
| NorESM1-Happi Coupled 30 years | Hist1 | 263.84 | 291.50 | 1.241 | 2.342 | 13.61 | 8.511 |
|  | RCP2.6 – Hist1 | +1.98 | +1.63 | +0.047 | +0.109 | -1.56 | -1.77 |
|  | RCP4.5 – Hist1 | +3.19 | +2.85 | +0.072 | +0.127 | -2.29 | -2.70 |
|  | RCP4.5 – RCP2.6 | +1.21 | +1.22 | +0.025 | +0.018 | -0.73 | -0.93 |

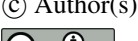


**Figures**

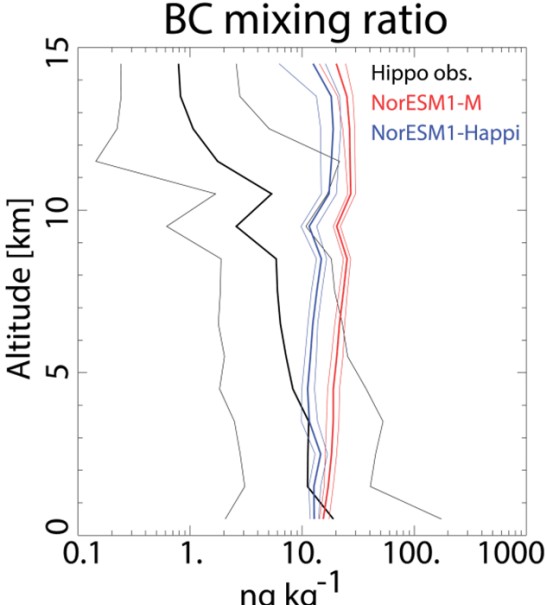

**Figure 1**. The modeled vertical profile of black carbon (BC) aerosols for AMIP-runs for PD (2006-
2015) with NorESM1-Happi (blue) and the fully coupled runs using RCP8.6 for the same period with
NorESM1-M (red). Thin lines indicate the inter-annual spread. Black lines are median (thick) and
ranges (thin) based on observations from the HIPPO campaign, with more than 700 vertical profiles
during 2009-2011 from 0.3 to 14 km altitude, mainly over the Pacific Ocean between 80ºN and 67ºS
(Schwartz et al., 2010 and 2013).

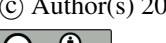


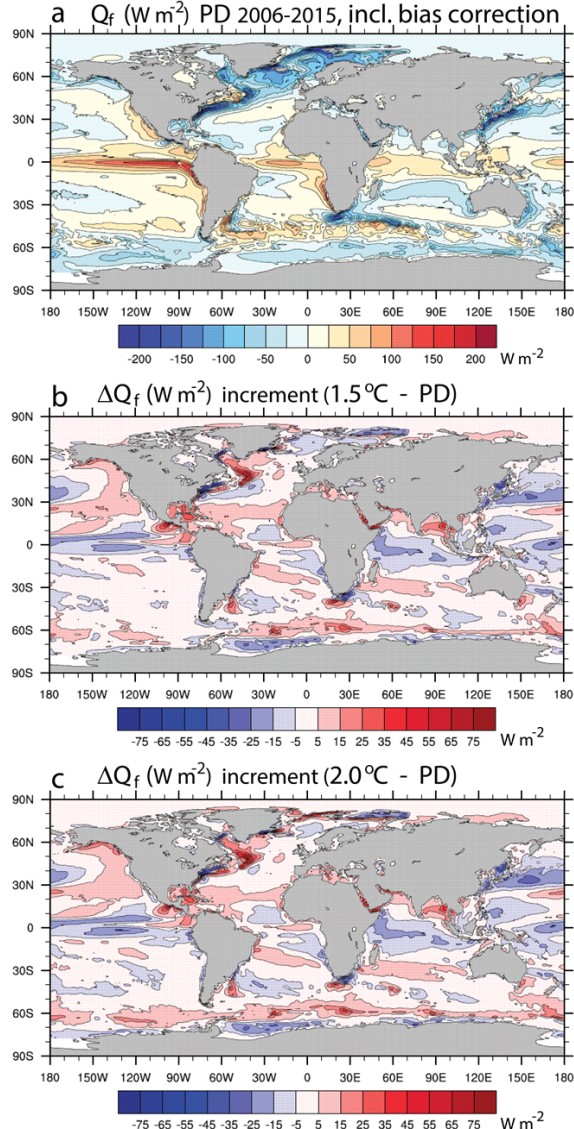

**Figure 2**. The annual average ocean net heat-flux $Q_f$ calibrated for the slab ocean model, NorESM1-HappiSO to maintain a stable climate with a present-day, 2006-2015 (PD), observed SST-field under vanishing TOA net heat-flux (a). The change in $Q_f$ from PD to a 1.5°C world (b) and from PD to a 2.0°C world (c), ensuring that the NorESM1-HappiSO produces a stable global climate comparable to the HAPPI AMIP runs with NorESM1-Happi without precribing the SSTs. Unit: W m$^{-2}$, negative values contribute to increasing SST (Eq. 1). The global averages are -1.32 W m$^{-2}$ (PD), -1.94 W m$^{-2}$ (1.5°C), and -1.71 W m$^{-2}$ (2.0°C).



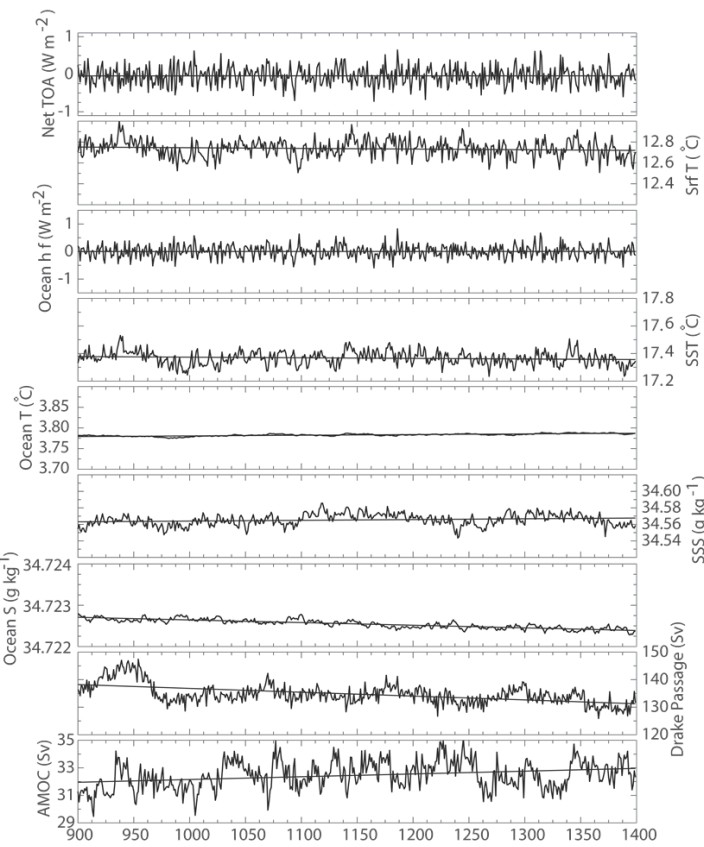

**Figure 3**. Annual mean time series between years 900 and 1400 from NorESM1-Happi piControl of, from top, net radiation at the Top of the Model Atmosphere, TOA, with positive values indicating warming of the atmosphere (mean value is -0.042 W m$^{-2}$); near surface air temperature (mean value is 12.74°C); net heat flux into the ocean/sea (positive value means ocean warming, mean value is +0.004 Wm$^{-2}$); SST (mean value is 17.37°C); volume-averaged ocean temperature (mean value is 3.78°C); SSS (mean value is 34.57 g kg$^{-1}$); volume-averaged ocean salinity ( mean value is 34.72 g kg$^{-1}$), net volume transport through the Drake Passage (mean value is 135 Sv), and the strength of AMOC at 26.5°N (mean value is 32.4 Sv). The black dashed lines in the two heat flux panels show the zero value, whereas the solid black lines in the other panels show the linear trends for years 900–1400. The figure can be compared with Fig. 2 in Bentsen at el. (2013), except that the mean value of the TOA net radiation in that paper should be corrected to +0.043 W m$^{-2}$.

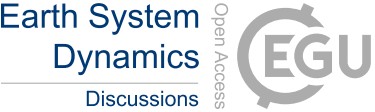

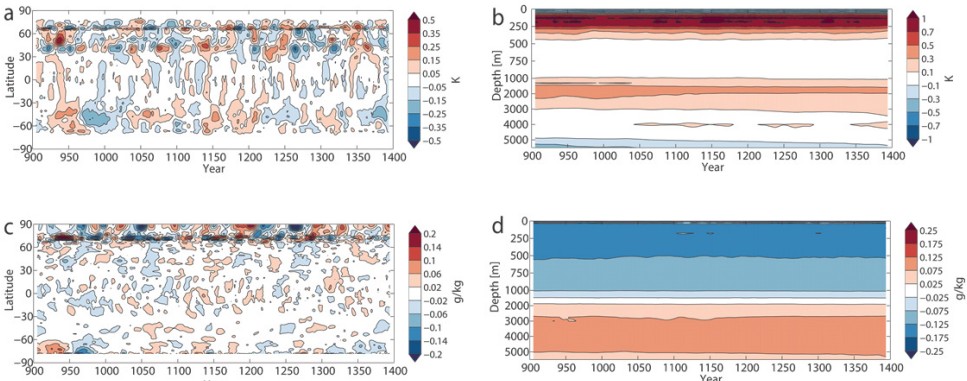

**Figure 4**. Latitude–time Hovmöller diagrams of (a) annual, zonal mean SST (K) and (c) SSS (g kg$^{-1}$) where the corresponding zonal time means have been subtracted, and depth–time Hovmöller diagrams of (b) global mean ocean potential temperature (K) and (d) salinity (g kg$^{-1}$) presented as anomalies compared to World Ocean Atlas 2009 (WOA09; Locarnini et al., 2010; Antonov et al., 2010) annual mean potential temperature and salinity. All panels are based on years 900–1400 of NorESM1-Happi piControl, filtered with a 10-year running mean. Note the non-linear depth co-ordinate in (b) and (d).



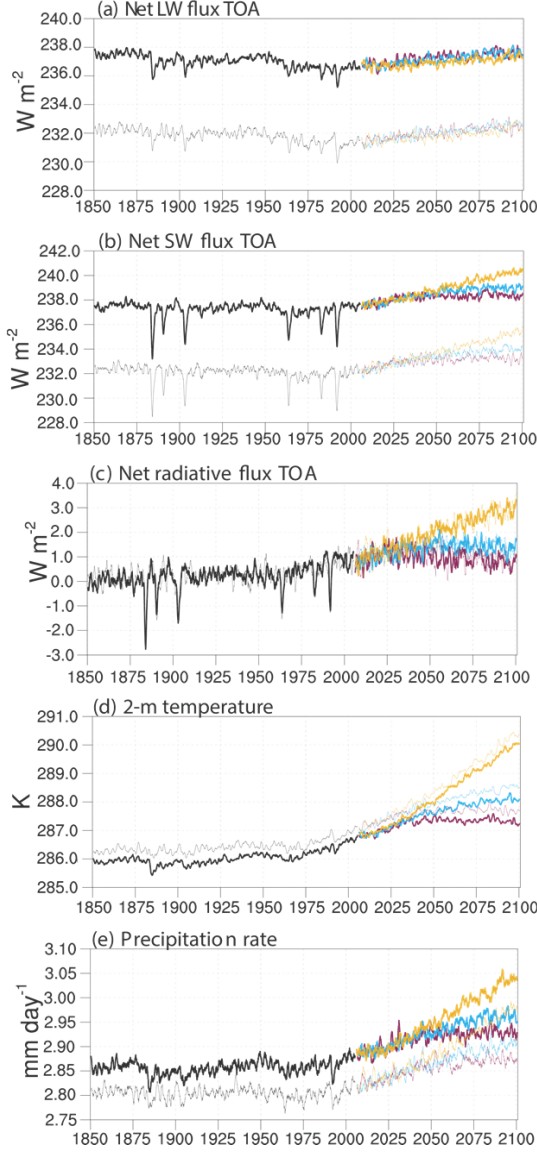

**Figure 5**. From the top panel and downwards, the figure shows the net global long-wave (positive upwards), short-wave (positive downwards), and total (positive downwards) radiative flux at the top of the model atmosphere (TOA) during the NorESM1-Happi coupled simulations for 1850 to 2100. The next two panels show diagrams for the global air temperature at reference height (2m) and monthly averaged precipitation rate. Black: Hist1; green: RCP2.6; blue: RCP4.5; red: RCP8.5. The curves for NorESM1-M (Iversen et al. 2013) are included as thinner lines for comparison.



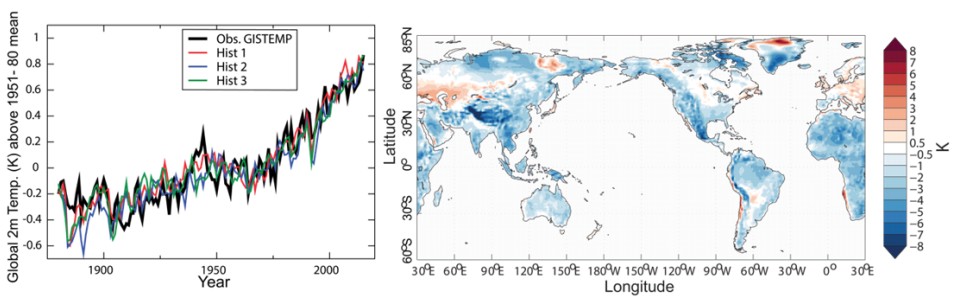

**Figure 6**. NorESM1-Happi global reference height air temperature compared with the NASA GISS
global temperature record (Hansen et al., 2010) over the historical period 1850-2015 (left), and
comparison of the reference height temperatures over continents for 1976-2005 with the CRU TS3.1
(Mitchell and Jones, 2005) observational data for the same period (right). Global bias is -1.45 K with a
RMSE of 2.40 K  (compared to -0.88 K and 2.38 K for NorESM1-M, see Table 3). In (a), three model-
calculated ensemble members are presented, while in (b) only the first ensemble member is shown.

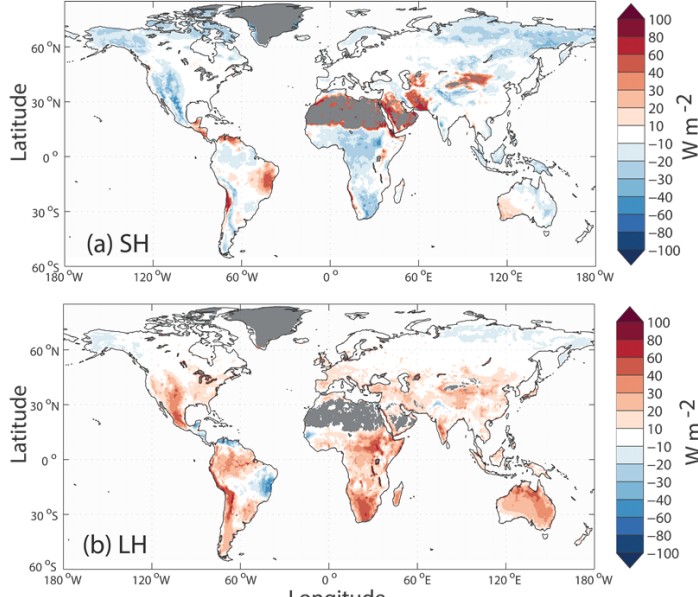

**Figure 7**. The difference between NorESM1-Happi and FLUXNET-MTE (Jung et al., 2011) fluxes of
sensible heat (a) and latent heat (b). The NorESM1-Happi fluxes are means for the years 1976–2005
of the Hist1 experiment, and the FLUXNET-MTE fluxes are means for the years 1982–2005. Areas
with missing observations are shaded with dark grey colour.





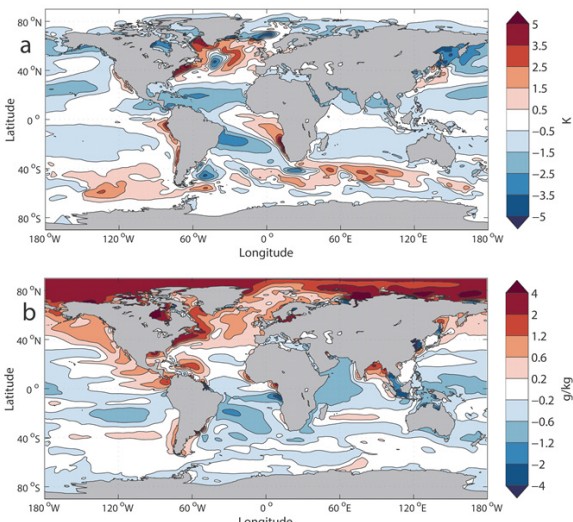

**Figure 8**. Differences between model calculated fields and observation-based data sets for SST (K) (a)
and SSS (g kg$^{-1}$) obtained from WOA09 (Locarnini et al., 2010; Antonov et al., 2010).

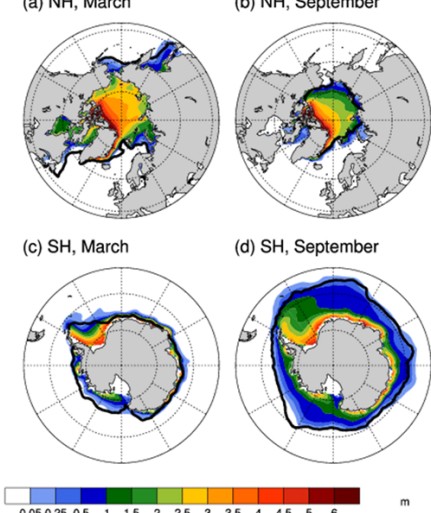

**Figure 9**. Mean sea-ice thickness (m) over years 1976–2005 of the NorESM1-Happi Hist1 experiment
for both hemispheres and for (a, c) March and (b, d) September. The solid black line shows the 15%
monthly sea ice concentration from the OSI-SAF reprocessed data set (OSI-SAF, 2017).



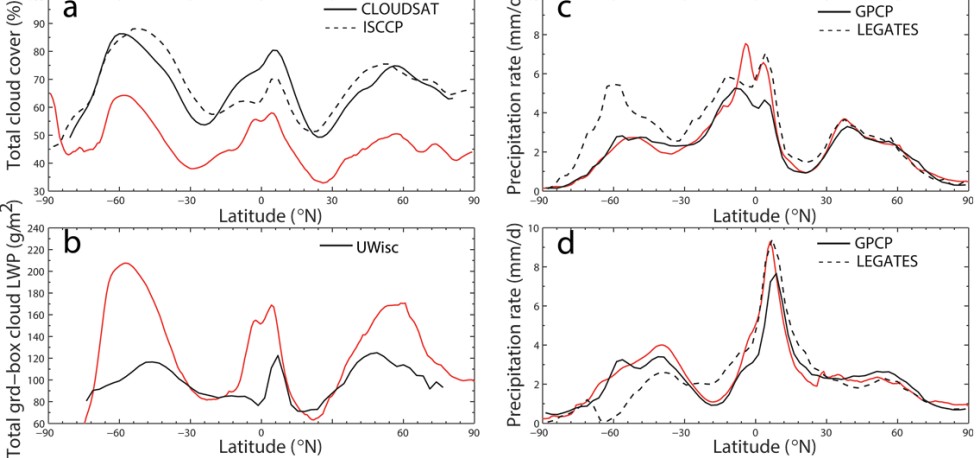

**Figure 10**. (a) Zonally averaged total cloud fraction (%) of NorESM1-Happi compared to ISCCP D2-retrievals 1983–2001(Rossow and Schiffer, 1999; Rossow and Dueñas, 2004) and Cloudsat radar and lidar retrievals from September 2006 – December 2010 (L'Ecuyer et al., 2008). **(b)** Zonally averaged total liquid water path (g m$^{-2}$) of NorESM1-Happi compared to UWisc retrievals over oceans for the period 1988–2008 (O'Dell et al., 2008). **(c)** Zonally averaged boreal winter (DJF) estimated annual precipitation of NorESM1-Happi compared to the data from GPCP (Adler et al., 2003) and Legates (Spencer, 1993; Legates and Willmott, 1990), and **(d)** the same for boreal summer (JJA). NorESM1-Happi values shown in red are means for the years 1976–2005 of the Hist1 experiment.

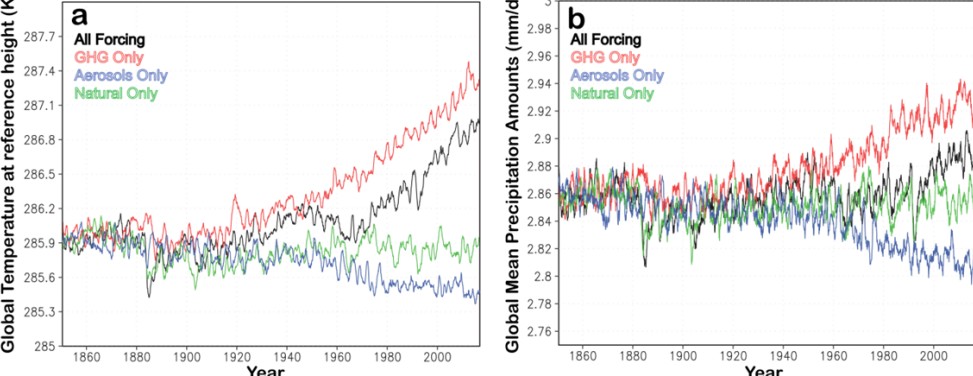

**Figure 11**. NorESM1-Happi global reference height air temperatures (a) and global mean diurnal precipitation amounts (b) over the historical period 1850-2015 for all and selected single forcings as well as the total forcing experiments for Hist1.



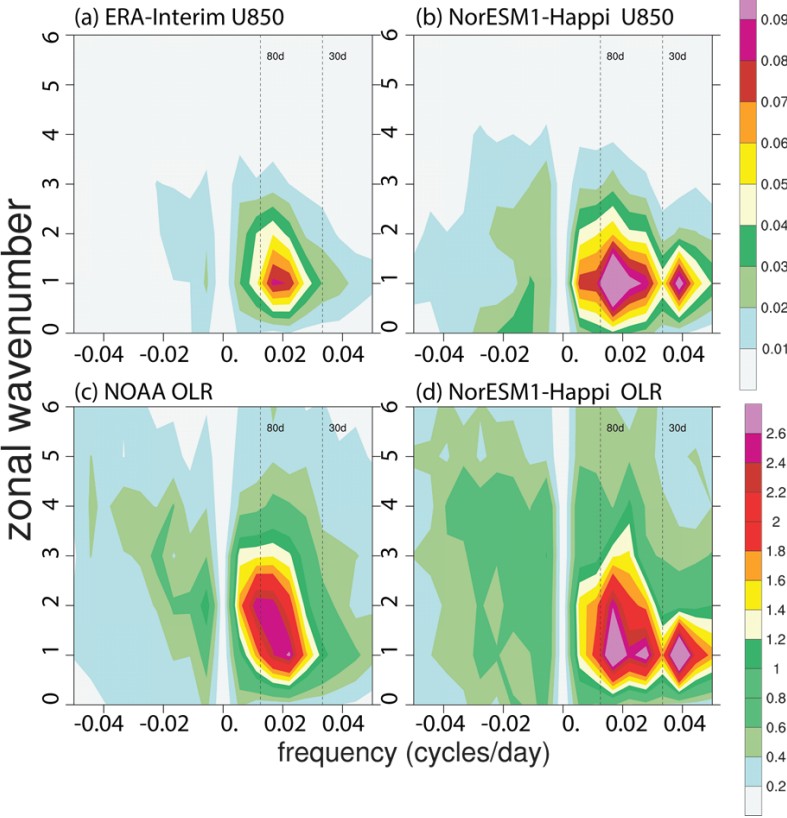

**Figure 12**. November–April wavenumber–frequency spectra of $10^o$ S – $10^o$ N averaged daily zonal 850 hPa winds of (a) ERA-Interim (1979–2008) (Dee et al., 2011) and (b) NorESM1-Happi (1976–2005), and daily outgoing long-wave radiation (OLR) of (c) NOAA satellite (1979–2008) and (d) NorESM1-Happi (1976–2005). Individual spectra were calculated for each year and then averaged over all years of data. Only the climatological seasonal cycle and time-mean for each November–April segment were removed before calculation of the spectra. Units for the zonal wind (OLR) are $m^{-2}$ $s^{-2}$ (W $m^2$ $s^{-1}$) per frequency interval per wavenumber interval. The band-width is 180 $day^{-1}$.





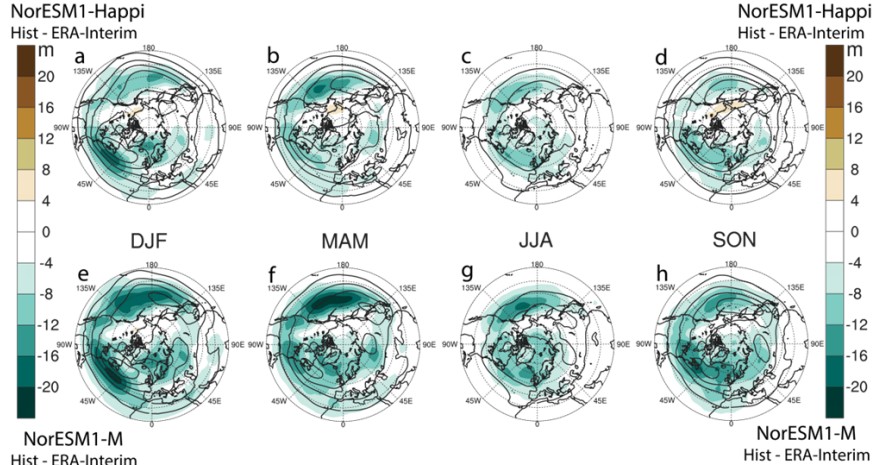

**Figure 13**. Estimated seasonal biases (coloured, unit m) of the NH extra-tropical cyclone activity, calculated as the standard deviation of the band-pass time-filtered geopotential height at 500hPa in the Hist ensembles (1976-2005) from NorESM1-Happi (a-d) and NorESM1-M (e-h) relative to the cyclone activity from ERA-Interim (1979-2008, Dee et al., 2011). Continuous lines show the Hist cyclone activity for the respective model. Cyclone activity is estimated by applying a 2.5 - 6 days band-pass time filter to the 500 hPa geopotential height before taking the standard deviation.

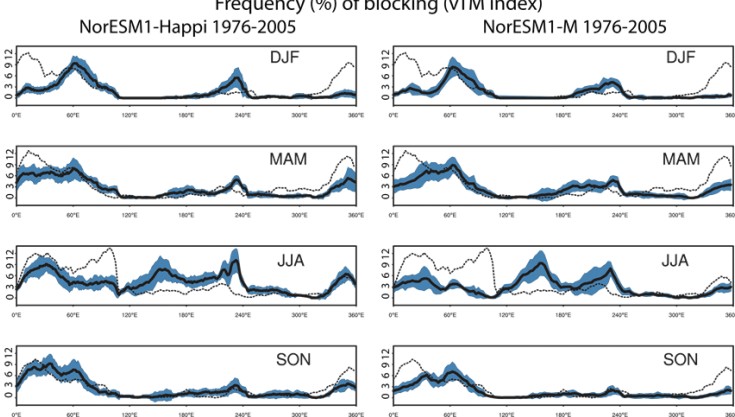

**Figure 14**. NH blocking frequency (%) for three Hist ensemble-members for NorESM1-Happi (left) and NorESM1-M (right) over the years 1976-2005 are presented. The solid black curves represent the ensemble mean, the blue shading is the ensemble spread (one standard deviation), and the dotted black curves are for the ERA-Interim data 1979-2008 (Dee et al., 2011). The seasonal occurrence of blocking is based on the 500 hPa vTM-index (Tibaldi and Molteni, 1990) relative to the latitudes of the average position of the westerlies by season (Pelly and Hoskins, 2002).




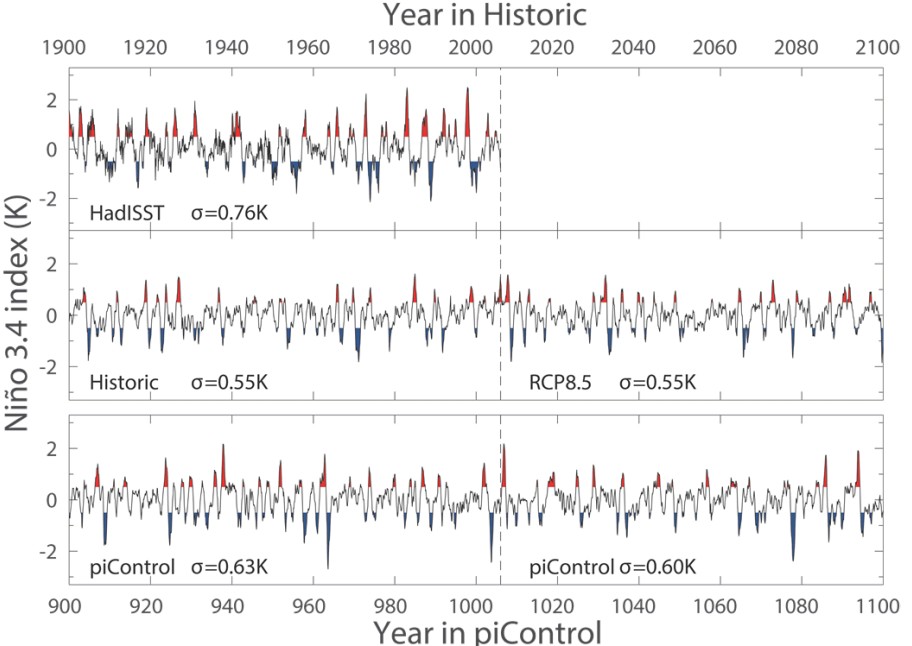

**Figure 15**. The time series of de-trended monthly SST anomalies of the NINO3.4 region (5°S-5°N;
170°W-120°W). The anomalies are found by subtracting the monthly means for the whole time series.
Red (blue) colours indicate that anomalies are larger (smaller) than +0.4 K (-0.4 K), see Trenberth
(1997) for recommendations. The upper time series shows Hadley Centre Sea Ice and SST data set
(HadISST; Rayner et al., 2003) for years 1900-2005; the middle time series consist of NorESM1-
Happi Hist1 for years 1900-2005 continued with NorESM1-M RCP8.5 for years 2006-2100; and the
lower time series displays NorESM1-Happi piControl for years 900-1100.





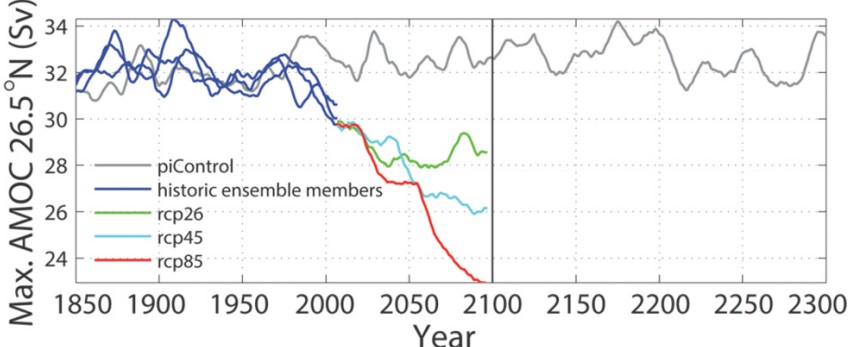

**Figure 16**. Decadal moving averages of the annual max AMOC at 26.5°N from the NorESM1-Happi
simulations for 1850 to 2100 are presented. Black represents the piControl, blue the 1850-2005 Hist1,
2 and 3; green the RCP2.6, turquoise the RCP4.5, and red the RCP8.5 projections over 2005-2100.
The grey curve is from the piControl

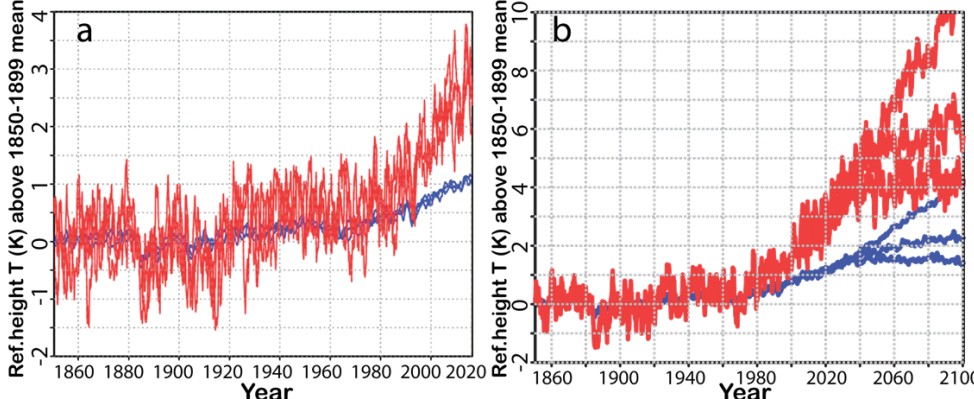

**Figure 17**. Simulated development in temperature at reference height with NorESM1-Happi relative
to the 1850–1899 average for the globe (red) north of 65° N (blue), i.e. ca. 4.7% of the global area.
Three historical ensemble members from 1850 to 2005 extended to 2015 with RCP8.5 are shown in
(a), and Hist1 until 2005 followed by a range defined by the three RCP scenario projections up to 2100
are shown in (b).



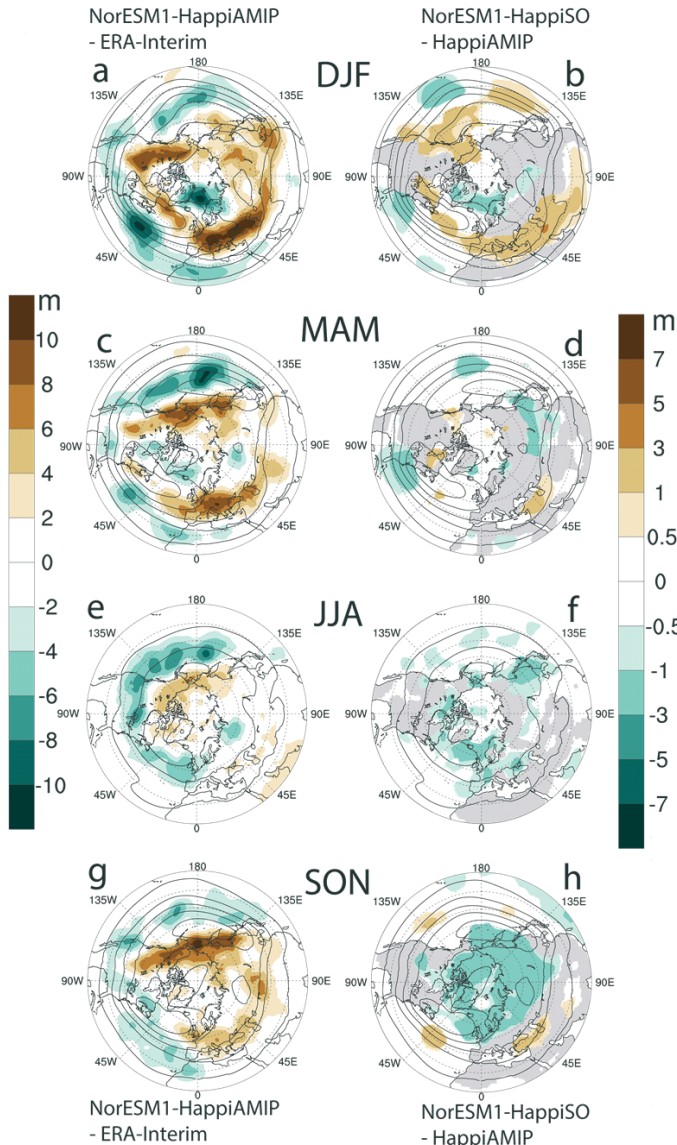

**Figure 18**. NH extra-tropical cyclone activity for PD (2006-2015). Seasonal results from the AMIP-ensemble for HAPPI with the NorESM1-Happi model compared to ERA-Interim data for the same decade (Dee et al., 2011) are shown as coloured shadings in a, c, e, and g. Differences between the results of the slab-ocean runs with NorESM1-HappiSO and the AMIP-ensemble are shown for each season as coloured shadings in b, d, f, and h. The black iso-lines are the diagnosed cyclone activity in the AMIP-ensemble (left) and SO ensemble (right) for each season. The shadings are shown where differences are statistically significant at the 5% level according to the Welch t-test.



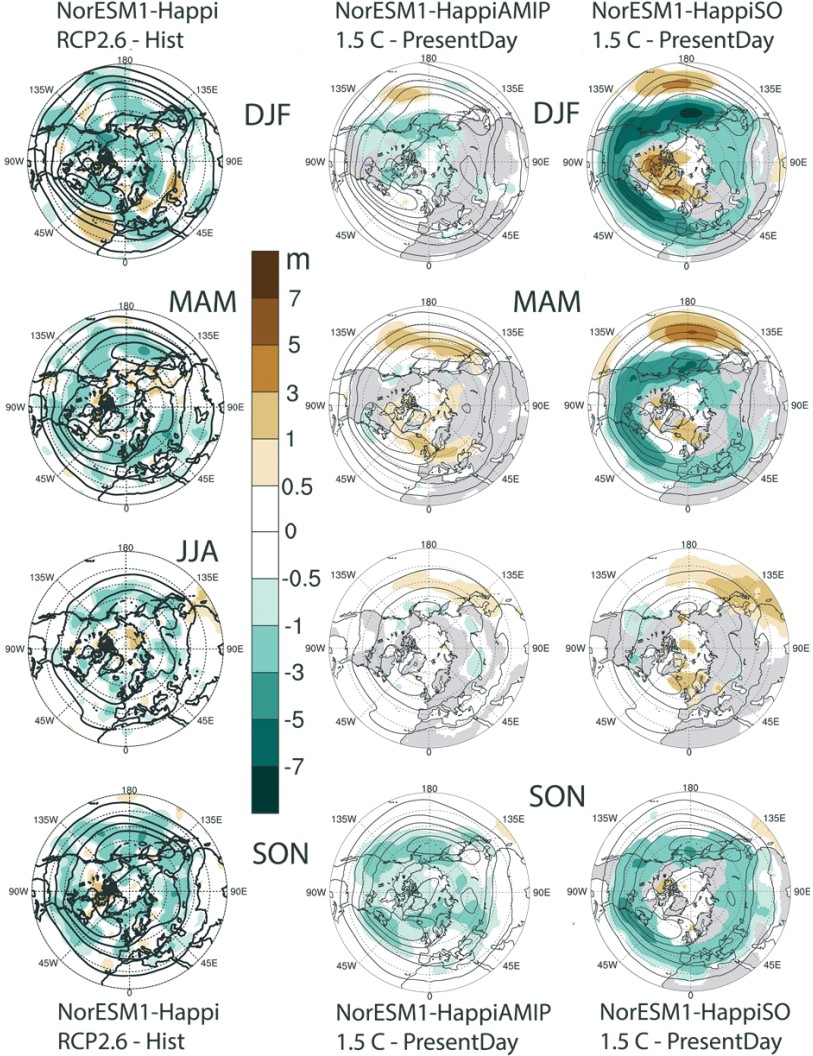

**Figure 19**. Coloured shadings show changes in the seasonal NH extra-tropical cyclone activity from
the Hist (1976-2005) to the RCP2.6 projection (2071-2100) with NorESM1-Happi (left column), and
from PD (2006-2015) to a 1.5°C warmer world estimated by the AMIP-ensemble from NorESM1-
Happi (middle column) and the slab ocean experiment with NorESM1-HappiSO (right column). The
black iso-lines are the diagnosed cyclone activity for Hist (left) or PD (middle and right) estimated
from the respective model simulations for each season. The shadings are shown where the differences
are statistically significant at the 5% level according to the Welch t-test.



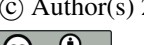

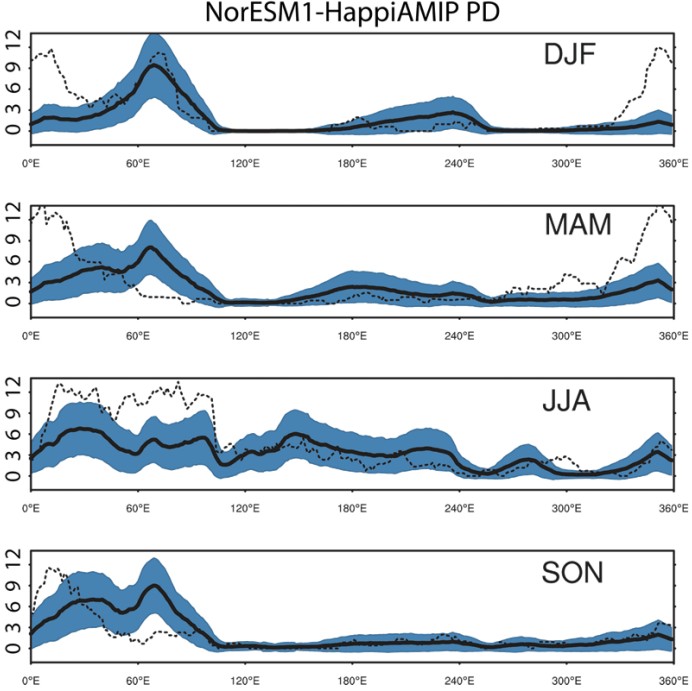

**Figure 20.** Blocking frequency of occurrence for the ensemble of NorESM1-Happi AMIP runs for PD (2006-2015). The solid black curve is the ensemble mean, while the blue shading shows the standard deviation in the ensemble of 10-year long runs around the ensemble mean. The dashed black lines show results for ERA-Interim over the same decade. Results for the 265 years with the NorESM1-HappiSO are almost identical to the results shown for the NorESM1-Happi AMIP runs.





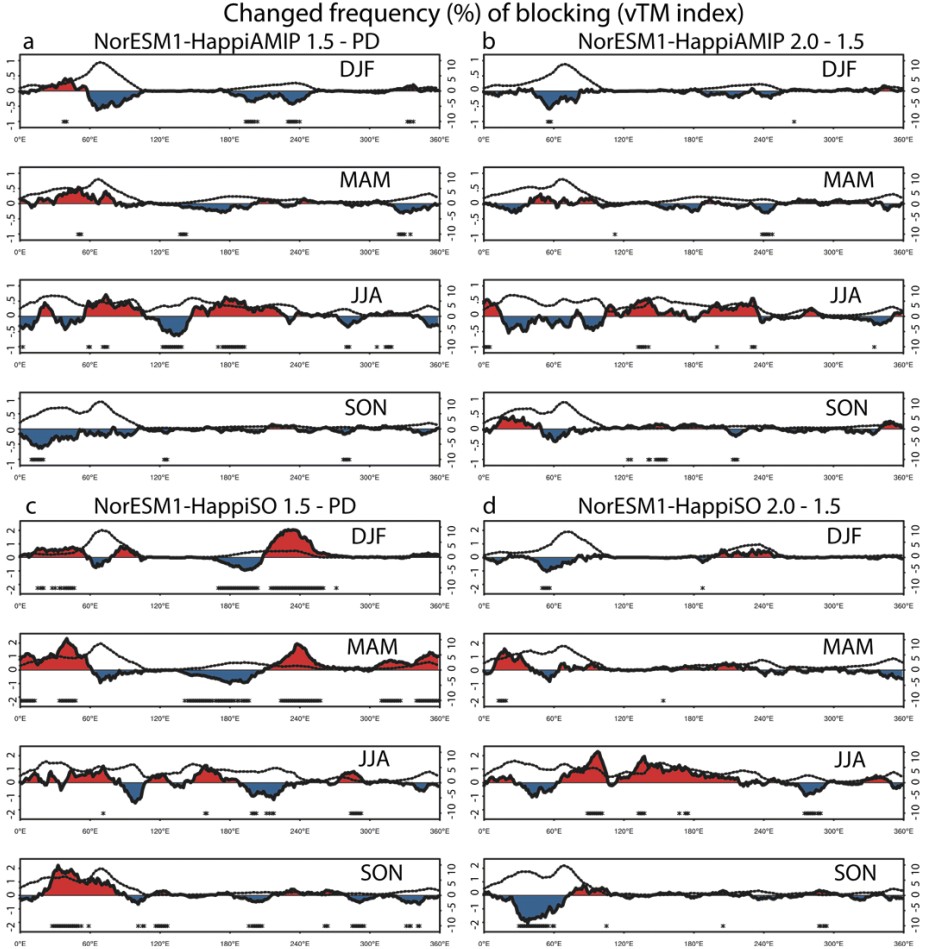

**Figure 21.** (a) Change in blocking frequency for the NorESM1-Happi AMIP ensemble experiment for
a 1.5 °C world compared to PD 2006-2015. The solid black line shows the ensemble mean difference
with blue shading indicating reduced blocking occurrence and red indicating increased. The dashed
black line shows the estimated PD climatology for reference. (b) Change in blocking frequency for
NorESM1-HappiAMIP ensemble for an additional 0.5°C global warming. The solid black line shows
the ensemble mean difference between the 2.0°C and the 1.5°C experiment, with blue and red shading
to indicate reduced and increased occurrence respectively. The dashed black line shows the 1.5°C
experiment climatology for reference. The diagrams (c) and (d) are similar results for the slab-ocean
model NorESM1-HappiSO. Dots mark differences that are statistically significant at the 5% level
according to the Welch t-test. *Note*: the y-axis to the left is for the colored difference, while that to the
right is for the climatological occurrence curves.



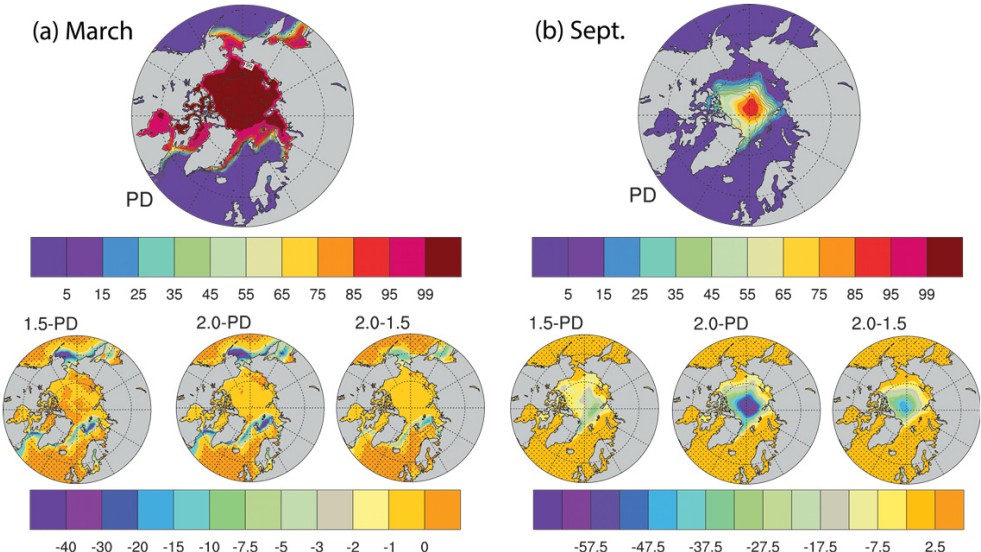

**Figure 22**. Calculated NH ice-concentrations (% of grid square area) for March (a) and September (b)
in the NorESM1-HappiSO model averaged over 265 years after 45 years of spin-up. PD are results for
2006-2015 together with observational estimates shown as solid black lines (OSI-SAF, 2017).
Increments in the ice-concentrations from PD to a 1.5°C warmer and a 2.0°C warmer world
respectively, and for the difference between the latter, are shown below. Differences are not
significant on the 5% level in areas marked with dots, according to the Mann-Whitney U test assuming
that different decades are un-correlated.

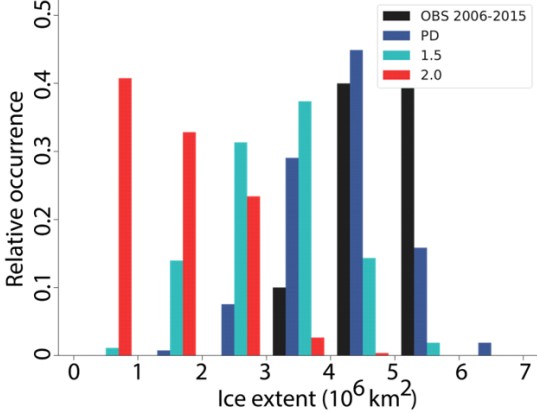

**Figure 23**. The relative occurrence of NH sea-ice extent counted in classes of width $1.\times10^6$ km$^2$ are
presented for the simulated PD climate with NorESM1-HappiSO (blue columns), to be compared with
observations (black columns; OSI-SAF, 2017), the 1.5°C warmer (turquoise columns), and the 2.0°C
warmer world (red columns) relative to the 1850 pre-industrial climate.