# Peer review of "Arctic amplification under global warming of 1.5 °C and 2 °C in NorESM1-Happi"

_Earth System Dynamics, 2017_

## Referee Comment (RC1) · Anonymous Referee #1 · 18 Jan 2018

This manuscript evaluates the quality of climate simulations with the NorESM1-Happi model, which is a slightly modified version of NorESM1-M (which was used in CMIP5). In addition, the authors discuss simulations with NorESM1-Happi that attempt to quantify the differential climate impacts between 1.5 and 2.0 degree global mean warming. Finally, the authors attempt to make inferences on the importance of coupled atmosphere-ocean-sea ice feedbacks in 1.5 and 2.0 degree global mean warming worlds, by comparison of AGCM-only and slab-ocean (SO) versions of NorESM1-Happi. Unfortunately I have major concerns and cannot recommend publication of this manuscript at this time, due to lack of clarity, lack of focus and the subject matter being potentially out of scope for ESD.

[Figure]

General comments:

1) While the focus of the manuscript is claimed to be the differential climate impacts of 1.5 and 2.0 degree global mean warming, it feels more like a model evaluation paper. Most text and figures are dedicated to the evaluation of NorESM1-Happi compared to the CMIP5 model NorESM1-M. While this is a useful and necessary exercise, I am not sure if ESD it the right platform to report on this.

2) Often times, there is lack of clarity in the text. Perhaps the first-author is not a native English speaker, but this should be addressed as it makes it challenging to follow the discussion. Examples: p.3 l. 34 ('AA has been calculated with a strong response?), p.11 l. 11 ('in reality?'), p. 11, l. 24 ('inaccurately calculated': was the calculation inaccurate or wrong?), p. 13, l. 6 ('calculated time-developments'), p. 14, l. 23 and Fig. 9 legend ('extension' –> 'edge'), p. 17 l. 29 ('describe' –> reproduce?'). Section 5.2 is really hard to follow: p. 22, l. 10 ('enhanced with the SO model compared to the AMIP': not sure what this means), p. 22, l. 11 ('latter': not sure what this refers to). p. 22 l. 15 (probably the tendency...: grammatically incorrect). p. 22, l 17-18 (unclear)

3) The presentation should be improved. There are way too many numbers listed in the tables (only a small fraction is discussed in text). This is overwhelming and makes it hard for the reader to know which are the most relevant ones. I also suggest to have a more consistent lay-out: in evaluation of NorESM-happi, sometimes NorESM-M results are included, and sometimes not (e.g. Figs. 10 and 12). I suggest to always include NorESM-M results for consistency. Also, the labelling in Fig. 13 is confusing, the x-axis in Fig. 14 is not legible (too small), and the different rcp's in Fig. 17b should have distinct colours.

4) From the text I cannot derive what the major scientific advance is from the comparison between the AMIP and SO results. Aren't the SO sea ice and SST fields meant to mimic the AMIP sea ice and SST fields? Does the fact that there are differences between the SO and AMIP sea ice and SST fields mean that the 'target' SST and sea

ice fields are not achieved in the SO runs? I'm concerned that this mismatch prevent a clean comparison of the climate impacts in the SO model versus the AMIP runs.

Other comments:

1) p. 3, l. 5: I'm not familiar with the term 'temperature ceiling', and find it a bit misleading. Better to use 'temperature target', as the temperature references in the Paris Agreement are generally believed to apply to long-term averages, and not to the maximum as the word 'ceiling' might suggest (see: doi: 10.1002/2017GL075612)

2) Fig. 1: For a cleaner comparison between models and observations I suggest to sample the model only at the locations where and times when observations were made.

3) p. 7, line 9: Do you really mean Fig. 11?

4) p. 8, l. 6-8: 'changes in the major elements ...': unclear, please elaborate.

5) p. 10, l. 21: what is the ACCESS version of the model? Please keep naming conventions/references to model versions consistent to avoid confusion

6) p. 12: maybe I missed it but did you discuss the trends in the AMOC and Drake Passage transport? These seem to be the quantities with the largest drift

7) p. 20, l. 21-22: Very confusing to first list the numbers in the brackets and not mentioning what they mean until 5-10 sentences thereafter.

8) p. 21, l. 1-10: I don't think this discussion is accurate. Sanderson et al. (2017) used an emulator to construct emissions pathways that would lead to a 1.5 and 2.0 degree warmer world. Those emissions pathways were then prescribed to a coupled atmosphere-ocean model. This is not a 'simplified method' as the authors suggest. Also, lines 2-4 are a bit misleading as well. It is now well established that the equilibrium climate response is determined by the accumulated carbon emissions. This implies that, in reality and in climate models that include a carbon cycle, switching off emissions

would lead to a rapid stabilization of global mean surface temperature (the delay in warming associated with ocean thermal inertia would be balanced by a decrease in greenhouse gas concentrations, see e.g. doi:10.1038/ngeo1047). My point is that it may not be as hard as the authors suggest to employ fully coupled atmosphere-ocean models to quantify the climate impacts of 1.5 and 2.0 degree global warming.

9) p. 21, l. 13: which 'forcing data', please specify

10) p. 21, l. 31: please give evidence that after 45 years a new equilibrium is indeed reached

11) p. 22, l. 6: What is meant with 'single projections'?

12) p. 22, l. 7: Table 7 does not show that the temperature targets (1.5 and 2.0 degrees above pre-industrial) are hit, but only show the temperatures relative to PD. What are the temperature relative to preindustrial?

13) As noted above, I found section 5.2 really hard to follow. What is particularly confusing is p. 22, l. 25-34. I think the sea ice area in the AMIP runs are compared to observations. Questions I have here are: 1) why is this comparison not presented earlier, 2) why are there any differences if the prescribed data is based on observations? 3) why is sea ice extent in Table 4 compared with sea ice area in Table 8?

---

## Referee Comment (RC2) · Anonymous Referee #2 · 30 Jan 2018

Review of Iversen et al "The NorESM1-Happi used for evaluating differences between a global warming of 1.5C and 2C, and the role of Arctic Amplification"

Summary

The authors look at a range of different versions of the NorESM1 model, and consider how those models hold up against reanalysis. They consider changes in many modes of variability, specifically related to key regional changes. Overall the paper was not what I expected, from the title I expected the paper would be about NorESM1-Happi, Paris Agreement and Arctic Amplification. Very little of this was even mentioned until Figure 17! As it stands the paper is $\frac{3}{4}$ a model description/validation paper, and $\frac{1}{4}$ a

science paper. The science is completely lost due to the first part. Due to this, and a number of other major concerns, I recommend substantial corrections.

Major concerns

1. In my view, the paper needs to be split into two. A paper focussing on the Arctic Amplification differences under Paris would be very welcome. So one suggestion is to put everything up to Figure 17 in online material, and just start the paper from there. Any reader that comes to this paper due to the title will be otherwise be completely lost in details of various models, and it will not be a productive read for them. More material would be needed for the science part though (see comments below). Alternatively, you could make this a model development only paper.

2. The title makes it seem that NorESM1-Happi is the main model here, but actually it is not, the SO version is used the most, and the –M and –AMIP versions are used equally as much. I often got confused about which one was being used, as the paper jumped around a fair bit. It was not until half way through that I realised that the Happi version of the model did not have prescribed SSTs (as HAPPI is synonymous with prescribed SSTs).

3. I was hoping to see more of a connection to Arctic Amplification here. Such as more of a focus on latitude temperature gradients, changes in wave characteristics associated with this, and then relating this to blocking etc. This link was missing, and AA just seemed to be a 'hot topic' term. The authors should look at the recent work by Screen on this topic.

4. A more comprehensive analysis seems to have already been done by the authors, in Li et al, (https://www.earth-syst-dynam-discuss.net/esd-2017-107/). Can the authors highlight what their study adds?

5. I was very surprised by some of the differences between the SO model and the AMIP model. Surely the AMIP model will have smaller biases that the SO model (e.g.

[Figure]

some Scaife papers could be referenced). It is not always clear that this is the case.

At this stage, I am not sure minor comments are useful.

---

## Author Response (AR1)

We want to thank the reviewers for their their constructive suggestions and comments. We have responded to their points below. In blue are the are the original responses suggesting how to revise the manuscript, and in red are our final responses, explaining what was done in the revised version of the manuscript.

**Anonymous Referee #1**

Received and published: 18 January 2018

This manuscript evaluates the quality of climate simulations with the NorESM1-Happi model, which is a slightly modified version of NorESM1-M (which was used in CMIP5). In addition, the authors discuss simulations with NorESM1-Happi that attempt to quantify the differential climate impacts between 1.5 and 2.0 degree global mean warming. Finally, the authors attempt to make inferences on the importance of coupled atmosphere-ocean-sea ice feedbacks in 1.5 and 2.0 degree global mean warming worlds, by comparison of AGCM-only and slab-ocean (SO) versions of NorESM1-Happi.

Unfortunately I have major concerns and cannot recommend publication of this manuscript at this time, due to lack of clarity, lack of focus and the subject matter being potentially out of scope for ESD.

**Reply to overall response:**

We understand the concerns of Referee #1 (as well as Referee #2) about the paper's scope for ESD, as well as the apparent lack of focus. Before we propose how to deal with this, please take into account that the manuscript was actually submitted to be considered for publication in GMD. It was intended for inclusion under the special collection of articles on the Norwegian Earth System Model (NorESM). This paper's version of the NorESM (NorESM1-Happi) is an update of NorESM1-M used for CMIP5. The AGCM-version of the model has been used to contribute to multi-model investigations in the HAPPI project, with AMIP-type experiments that address differences between a 1.5 degrees and a 2.0 degrees warmer world than the pre-industrial (1850). Our intentions with the present paper, when submitting it to GMD, were

- 1. to validate the fully coupled NorESM1-Happi which employs the atmosphere and land components that are used in the experiment in the HAPPI project, e.g. by comparing its performance to NorESM1-M, and
- 2. to apply a slab-ocean version (NorESM1-HappiSO) which includes the sea-ice model from the fully coupled NorESM1-Happi, to investigate how the polar amplification of the temperature signal may change the modelled difference between a 1.5 and a 2.0 degree world. In addition, results from the RCP2.6 and RCP4.5 runs with NorESM1-Happi complement the discussions, although these scenarios are not targeting the two temperature increments per se.

Obviously, the visibility of point (2) has suffered due to the lengthy and detailed discussions

over the first 4 sections. Point (1) was emphasized in the manuscript when designing it for GMD rather than ESD. The topic editor of GMD who was assigned for the paper almost a month after the submission, chose to reject the paper for GMD and recommended to transfer it to ESD. With the publication constraints for papers to be referred to in the scheduled Special Report from IPCC in mind, we chose to follow the recommendation without much discussion, although we suspected that the profile of the manuscript could produce confusion with the ESD editors and reviewers.

Now, with both the referees' comments in mind as well as our own concern about the paper's

profile for ESD, we propose to restructure the manuscript as follows, hoping to achieve stronger focus, better clarity, and a more suitable profile for ESD.

- We propose to reduce the amount of material that documents the standard validation of NorESM1-Happi as a global climate model, mainly by compressing much of the information given in the present text into tables, and collect them in a separate set of "supplementary material". A considerably shortened text will still be kept in one sub-section of the main manuscript, where the model design is described. Some validation results that are relevant for the polar amplification topic will be kept in the main text (feedback analysis, extratropical storm-tracks and blocking, sea-ice and aspects concerning the freshwater cycle).
- A clearer presentation of the reasons for running the slab ocean experiments to complement the AMIP-type experiments will be given, and in particular the replacement of a prescribed sea-ice cover in the AMIP-runs with a fully dynamic-thermodynamic sea-ice in the slab ocean model runs.
- We are underway with a selection of simulations with the fully coupled NorESM1-Happi model, which target the 1.5 degree and the 2.0 degrees warmer world (compared to 1850 pre-industrial). This will better complement the AMIP and SO-experiments, with specific focus on polar amplification than the runs based on RCP2.6 and 4.5 that are now included.

All in all, this should considerably strengthen the paper's focus on polar amplification, while still including the necessary results in support of the model validation as a secondary item.

With this, we believe the paper will be suitable for ESD, and in particular to be included in the

special issue about The Earth system at a global warming of 1.5°C and 2.0°C ( https://www.earth-syst-dynam.net/special\_issue909.html).

We have made substantial changes to the paper. It has been restructured and now is now focused on how ocean and sea-ice feedbacks affect the response to 1.5 degree and 2.0 degree warming. Details are given below:

• Following the suggestion of reviewer 2, we have moved the standard validation of NorESM1-Happi against NorESM1-M, observations and reanalysis out of the main manuscript and into a Supplement. The material has furthermore been condensed

substantially. A very short summary of the most important results are kept in the model section in the main text. We now only include some validation results for fields that are relevant for the discussion in the main text. We have also shortened the section describing the fully coupled NorESM1-Happi.

- We have designed and carried out new simulations with the fully-coupled NorESM1-Happi for 1.5 degree and the 2.0 degrees warming targets (compared to 1850 pre-industrial). Results from these simulations are compared to results from the SO and AMIP experiments, and replace the results from RCP2.6 and RCP4.5 in relevant cases.
- We have updated the slab-ocean simulations using Q-fluxes that are are calibrated using SST increments (difference between the PD SSTs and the 1.5C and 2.0C warmer SSTs) from the fully-coupled runs instead of the SST increments from the AMIP forcing data. The reason is that the increments from the fully-coupled simulations are more consistent with the model climate.
- We have added a new section dedicated to describing the set-up of the 1.5 degree and 2.0 degree warming experiments carried out with the AMIP version, the SO version, and the fully coupled version of NorESM1-Happi. This new section should make the differences between the AMIP, SO, and fully-coupled experiment more clear, but also accounts for the motivation behind running the fully-coupled and the SO experiments to complement the AMIP-type experiments.
- In addition to adding results from the fully-coupled simulations to compliment the already existing analysis of the AMIP and SO experiments, we now also consider changes in the upper-level and lower tropospheric equator-to-pole temperature gradients and changes in the upper-level and lower-level extratropical storm tracks. We also discuss how these results compare with the multi-model analysis carried out by Li et al. (2018).

The manuscript is now more clearly focused on the role of ocean and sea-ice feedbacks and polar amplification, while still including the necessary results for model validation in a Supplement.

We believe the revised paper is now well suited for ESD.

**General comments:**

**1)**

While the focus of the manuscript is claimed to be the differential climate impacts of 1.5 and 2.0 degree global mean warming, it feels more like a model evaluation paper. Most text and figures are dedicated to the evaluation of NorESM1-Happi compared to the CMIP5 model NorESM1-M. While this is a useful and necessary exercise, I am not sure if ESD it the right platform to report on this.

**Reply:**

As mentioned in the overall response, the paper was written for and submitted to GMD, but the assigned topical editor thought it was better suited for ESD. We now realize that the title of the manuscript may have given the GMD topical editor the impression that the paper is not a model evaluation paper (our fault). Since ESD has a special issue on "The Earth system at a global warming of 1.5°C and 2.0°C", we decided to follow the suggestion of the GMD topical editor (even though we did expect that the ESD-referees probably would comment on this).

We are therefore prepared to restructure the paper, and propose to include major parts of the pure model validation in a "supplementary material", reduce its volume, and only keep a shorter discussion in a subsection of the main text. Instead, the discussion of the differences in climate between a 1.5 and a 2.0 degrees warmer world will be more prominent, with the role of polar amplification as the paper's focus.

See changes listed under reply to general comments above.

**2)**

Often times, there is lack of clarity in the text. Perhaps the first-author is not a native English speaker, but this should be addressed as it makes it challenging to follow the discussion. Examples:

**Reply:**

We intend to seek help to copyedit the updated (or the very final) version of the manuscript by a native English speaker.

The manuscript has been largely re-written, both to change the scope from a model validation paper to a science paper and to improve the clarity of the text. The Supplement and parts of the manuscript have been proofread by a native English speaker.

p.3 I. 34 ('AA has been calculated with a strong response?),

**Reply:**

The sentence should read:

Even for the remote and regionally localized forcing caused by reduced European sulphate aerosols since the 1980s, the strongest amplification of warming is found in the Arctic (Acosta Navarro et al., 2016).

This sentence has been rewritten. See p3, L25-26.

p.11 I. 11 ('in reality?'),

**Reply:** Replace by "already", and in addition modify the entire paragraph to read:

NorESM1-Happi is a version of NorESM1-M with relatively minor updates. The most radical difference is a doubling of the horizontal resolution in the atmosphere and land models. As NorESM1-M is already thoroughly documented through CMIP5, this paper presents only selected features of the updated NorESM1-Happi.

This section has been moved to the Supplement and rewritten.

p. 11, I. 24 ('inaccurately calculated': was the calculation inaccurate or wrong?),

**Reply:**

*"incorrectly" is the correct phrase.*

This part of the text has been moved to the Supplement and rewritten.

p. 13, I. 6 ('calculated time-developments'),

**Reply:**

We propose to amend the sentence (if it will be used in the new manuscript) to become:

*Fig.* 5 shows the simulated time evolution from 1850 to 2100 of some of the quantities in Table 2.

*Fig.* 5 has been moved to the supplement (now Fig. S8) and the text describing it has been modified.

**p. 14, l. 23 and Fig. 9 legend ('extension' -> 'edge'),**

**Reply:**

will be changed to "edge".

Comment: "ice extent" will replace "ice extension". Unfortunately, in a few places the ice extent has been confused with ice area. Ice area takes into account the fractional ice-cover in any grid cell, while the ice extent measures the size of the entire domain where sea-ice is present. This will be corrected.

The second sentence in the Fig. 9 legend should read:

The solid black line shows the sea-ice edge estimated as the 15% iso-line of monthly sea-ice concentration in the OSI-SAF reprocessed data set (OSI-SAF, 2017).

This part of the text has been rewritten and moved to the Supplement along with Fig. 9. The second sentence in the legend of Fig. 9 (now Fig. S4) now reads: The solid black line shows the climatological 15 % concentration line for the same period from the OSI-SAF reprocessed data set (OSI-SAF, 2017).

p. 17 I. 29 ('describe' -> reproduce?').

Reply:

agreed. "reproduced" is better.

We use "reproduced" in the revised text. This part of the text has been moved to the Supplement (Figure S13).

Section 5.2 is really hard to follow:

**Reply:**

We are sorry for the sloppy and confusing language many places. This section will be re-written and expanded in a new manuscript.

This section has been rewritten to accommodate the new results and for clarity.

p. 22, I. 10 ('enhanced with the SO model compared to the AMIP': not sure what this means),

**Reply:**

**see next point.**

p. 22, I. 11 ('latter': not sure what this refers to).

**Reply to both points:**

Sorry! Yes, this is confusing. The sentences should read:

The PAF is considerably larger in the Arctic than in the Antarctic. Furthermore, the SO model produces stronger AA (by 18%) than the AMIP model.

This section has been rewritten to accommodate the new results and for clarity.

p.22 I. 15 (probably the tendency...: grammatically incorrect).

**Reply:**

see next point.

This part of the text has been rewritten.

**p. 22, I 17-18 (unclear)**

**Reply to both points:** The following amended text is proposed for the entire paragraph:

The SO model has a tendency to produce a colder winter climate than the AMIP model, consistent with the systematic cold bias in the fully coupled system. The differences in summer climate between the SO and AMIP simulations are much smaller. The SO model generally simulates less reduction in sea ice area than is prescribed in the AMIP simulations, with the exception of the NH summer response for 2.0 degrees - PD. These differences are also reflected in the temperatures and precipitation over land.

**Reply:** Please see the detailed replies above.**

We suspect that missing clarity is also a consequence of the hectic situation caused by the deadline for submission on Nov. 1 st 2017 (for papers to be referred to in the IPCC's special report on 1.5 degrees). We will pay considerable attention to the clarity of language when updating the manuscript, and we intend to seek help by a native English speaker before finalization.

This part of the text has been rewritten to accommodate the new results and to make it more clear.

**3)**

The presentation should be improved. There are way too many numbers listed in the tables (only a small fraction is discussed in text). This is overwhelming and makes it hard for the reader to know which are the most relevant ones. I also suggest to have a more consistent lay-out: in evaluation of NorESM-happi, sometimes NorESM-M results are included, and sometimes not (e.g. Figs. 10 and 12). I suggest to always include NorESM-M results for consistency.

Also, the labelling in Fig. 13 is confusing, the x-axis in Fig. 14 is not legible (too small), and the different rcp's in Fig. 17b should have distinct colours.

Reply:

The many numbers are there to document basic properties in order to validate the model as a suitable tool for the study of processes in the climate system. We do not explicitly comment on every number in the text, although the overall model properties, e.g. concerning the simulation of a stable pre-industrial climate, climate sensitivity, and climate variability and change, are discussed. Such numbers are essential in a climate model validation. We understand, however, that a full model validation was not to be expected in an ESD-article, or from the title of the paper. We believe this will be considerably improved by moving many of the tables to a "supplementary material".

We think it can be a good idea to include NorESM1-M results in Figs. 10 and 12 as well, to the extent that they bring substantial additional information (there are already many figures). The details mentioned for Figs. 13, 14 and 17b will be taken into account. The figures will in any case be changed when re-organizing the manuscript.

To limit the number of figures in the Supplement, we prefer not to add versions of Figures 10 and 12 (now Figures S3 and S10) for NorESM1-M. We have added references to the figures in Bentsen et al. in the relevant captions to make it easier for the reader to locate the corresponding plots for NorESM1-M.

We have changed the labeling of Figure 13 (now Figure S11).

We have increased the font height of the labels for the x-axis in Figure 14 (now figure S12).

Figure Fig 17b has been updated and now show the time-evolution of the fully-coupled 1.5 degree and 2.0 degree warming experiments instead of the RCPs. We have kept the colors as we feel that adding more colors makes it more confusing.

**4)**

From the text I cannot derive what the major scientific advance is from the comparison between the AMIP and SO results. Aren't the SO sea ice and SST fields meant to mimic the AMIP sea ice and SST fields? Does the fact that there are differences between the SO and AMIP sea ice and SST fields mean that the 'target' SST and sea ice fields are not achieved in the SO runs? I'm concerned that this mismatch prevent a clean comparison of the climate impacts in the SO model versus the AMIP runs.

**Reply:**

We realize that the description of the differences between the SO-experiments and the AMIP-experiments is insufficient. We will expand the text accordingly as indicated below.

The referee is, quite understandably, concerned about the fact that the SO-model simulates sea-ice properties (and therefore also to some extent SSTs) which deviate considerably from the AMIP experiments, even though the calibration of the slab

ocean is constructed to mimic the prescribed ocean state used for the AMIP experiments.

Does this imply that the SO-model is profoundly wrong, and is not valid for the intended study of polar amplification? **The reply is no.** The explanation follows.

The SO model includes interactive sea-ice and snow processes which are missing in the AMIP-runs. In principle, the mean climatology for the SSTs used in the AMIP experiments should be reproduced by the SO-model experiments, provided the sea-ice properties (cover, thickness, concentration) are the same in the two setups. However, when calibrating the SO-model, we do not directly control the sea-ice properties. We define deep-ocean heat-fluxes (Q) in the SO in order to approximate the SSTs used in the AMIP experiments, but there is no similar relaxation of the sea-ice in the SO model towards the sea-ice which is prescribed in the AMIP runs. The Q-fluxes will indirectly constrain the SO-calculated sea-ice to some extent, but there is no reason why this sea-ice should be identical to the AMIP sea-ice.

We feel that this is a strength of the SO runs compared to the AMIP runs. The sea ice concentration prescribed in the present day AMIP experiment are based on satellite observations, but other properties of the ice and snow cover are more arbitrarily set as described in detail in Mitchell et al. (2017; doi:10.5194/gmd-10-571-2017). For example, the sea-ice thickness is 2 m over the entire Arctic sea ice extent and 1m in Antarctica, and the snow-cover is allowed to become unrealistically thick in many places. We will extend the description of this in an updated manuscript, thus clarifying why the SO-model with its more realistic feedback processes associated with sea-ice and snow cover, is useful.

The heat isolation and radiation properties of sea-ice thickness and snow cover are particularly important for e.g. the Arctic temperatures over extended winter seasons. In the SO-model, these properties are allowed to vary in accordance with physical processes represented in the sea-ice model, while the sea-ice thickness is constant in the AMIP runs. Hence, the SO-simulation should have a more realistic – or at least more physically based – representation of feedbacks that influence the Arctic amplification in ways not included in the AMIP runs.

Concerning SSTs, the SO-model should produce results close to the prescribed SST-fields used in AMIP, except in regions directly influenced by the sea-ice model results. By construction, the relaxation we use to define the Q-fluxes secures that the SO-model results hit the same global temperature targets as the AMIP runs. Deviations originate from the fact that the atmosphere in the SO-model "sees" a different sea-ice and snow cover than in the AMIP runs.

We will provide a comparison of the simulated sea-ice thickness changes in the 1.5C and 2.0C experiments that are underway with the fully coupled NorESM1-Happi model with those of the SO experiments, although the fully coupled experiments are not bias-corrected for present-day such as the SO-experiments.

**In conclusion**, and this will be better presented in an updated manuscript, the results from the SO-model should produce an Arctic amplification of the temperature response which differs from the AMIP-results. The differences are dominated by the changes in the sea-ice and snow cover in the SO-model with some regional influence on the SSTs. Otherwise, the contributions to AA from SST-feedbacks directly, should be included in the SST-fields prescribed for the AMIP experiments.

We have added a section describing the warming experiments carried out with the AMIP, SO, and fully-coupled version of NorESM1-Happi to make it more clear how the experiments with the different model versions compliment each other. We have also improved the section describing the SO model.

We now provide results for changes in sea ice for both the SO and the fully-coupled model.

**Other comments:**

**1) p. 3, l. 5:**

I'm not familiar with the term 'temperature ceiling', and find it a bit misleading. Better to use 'temperature target', as the temperature references in the Paris Agreement are generally believed to apply to long-term averages, and not to the maximum as the word 'ceiling' might suggest (see: doi: 10.1002/2017GL075612)

**Reply:**

We were concerned with the term "target". The target is in principle an upper bound of the global temperature change, not a target that one wishes to hit. However, if "ceiling" is a misleading word, we can change to "target", while also explaining that an under-shoot should not be considered a miss.

We do not use the term "temperature ceiling" in the revised manuscript.

2) Fig. 1: For a cleaner comparison between models and observations I suggest to sample the model only at the locations where and times when observations were made.

**Reply:**

In fact, model output was already sampled in grid volumes containing the observation points, and for dates in each of the ten years of the PD-period that coincide with the observation dates. Since the model is run in climate simulation mode without data-assimilation (or nudging), we cannot compare the exact observation times. We will update the figure legend to make this clear. The figure will be moved to the supplementary material section. We have updated the figure legend to make this clear. The figure has been moved to the Supplement (now Fig. S1).

3) p. 7, line 9: Do you really mean Fig. 11?

**Reply:**

*No, it should be Fig. 9 (now Fig. S4). This part of the text has been rewritten and moved to the caption of Fig. S4.*

4) p. 8, l. 6-8: 'changes in the major elements ...': unclear, please elaborate.

**Reply:**

More specifically: GHG, aerosols and land-use

This paragraph has been removed as we now dedicate a section to the set-up of the 1.5 degree and 2 degree warming experiments.

**5) p. 10, l. 21:**

what is the ACCESS version of the model? Please keep naming conventions/references to model versions consistent to avoid confusion

**Reply:**

Sorry about this. This sentence remained from an early version of the manuscript. It should read:

Before the piControl, the 1-degree version of the atmosphere and land model described in Seland and Debernard (2014) was spun up over 300 years, starting from....

We refer to the different model versions in a more consistent way in the revised manuscript.

**6) p. 12:**

maybe I missed it but did you discuss the trends in the AMOC and Drake Passage transport? These seem to be the quantities with the largest drift

**Reply:**

The trends in both quantities are mentioned on p12, I 3-11, and compared to the trends in the NorESM1-M model. They are not discussed in depth, and we will add a sentence in the manuscript stating that while the apparent drift in those indices indicates that the deep ocean is not in perfect balance, we do not think is a serious issue our study. The magnitude of the trends are comparable to those in NorESM1-M (Bentsen et al, 2013).

We have shortened this part of the text and moved it to the Supplement (caption of Table S3). We mention that the magnitude of the trends are comparable to those in NorESM1-M (Bentsen et al, 2013).

Otherwise, the strong AMOC leads to heating of the deep oceans and leaves less energy for surface heating and evaporation. We will add a figure in the planned supplementary material, which documents how heat penetrates into the deep ocean during simulated RCP scenario projections with NorESM1-Happi. On p13, I 14-30, we hypothesize a connection between the strong AMOC and the simulated speed of the atmospheric freshwater cycle, as well as the under-estimated cloud cover that can be related to the horizontal resolution of the atmosphere.

We have added figures in the manuscript showing the AMOC (Figure 4) and how heat penetrates into the deep ocean (Figure 3) in the fully-coupled simulations of the 1.5 degree and 2.0 degree warmer worlds.

**7) p. 20, l. 21-22:**

Very confusing to first list the numbers in the brackets and not mentioning what they mean until 5-10 sentences thereafter.

**Reply:**

Agreed. We will mention this in the beginning of the paragraph.

We have moved Table 6 and the paragraph discussing it to the Supplement (Table S6). The text has been rewritten for clarity and shortened.

p. 21, l. 1-10:

I don't think this discussion is accurate. Sanderson et al. (2017) used an emulator to construct emissions pathways that would lead to a 1.5 and 2.0 degree warmer world. Those emissions pathways were then prescribed to a coupled atmosphere-ocean model.

This is not a 'simplified method' as the authors suggest.

**Reply:**

Of course, this is wrongly stated in the paper and will be corrected.

**We have corrected this in the revised manuscript.**

Also, lines 2-4 are a bit misleading as well. It is now well established that the equilibrium climate response is determined by the accumulated carbon emissions. This implies that, in

reality and in climate models that include a carbon cycle, switching off emissions would lead to a rapid stabilization of global mean surface temperature (the delay in warming associated with ocean thermal inertia would be balanced by a decrease in greenhouse gas concentrations, see e.g. doi:10.1038/ngeo1047). My point is that it may not be as hard as the authors suggest to employ fully coupled atmosphere-ocean models to quantify the climate impacts of 1.5 and 2.0 degree global warming.

**Reply:**

**We appreciate the referee's point.**

We are now underway with some experiments with the fully coupled NorESM1-Happi model, targeting a 1.5 and a 2.0 degrees warmer world than the model simulated pre-industrial control run for 1850. These experiments are not emission-driven for GHGs, hence we do not base our calculations on the principle from Gillett et al. (2011, doi:10.1038/ngeo1047). Instead, we have designed GHG-based forcing with temperature targets based on amendments to RCP2.6 and 4.5, and our own estimates of the model's climate sensitivity.

We will not have time to produce an ensemble of model projections comparable to that of Sanderson et al (2017), but hopefully around 150 simulation years per temperature target. Statistics will replace those included for the RCP2.6 and RCP4.5 in the present manuscript, and thus better complement those of the SO- and AMIP-experiments.

We have conducted fully-coupled simulations for 1.5 and 2.0 degree warmer worlds. The set-up of the experiments is presented in Section 2.2 and the results are discussed alongside the results from the SO and AMIP experiments in the result sections.

9) p. 21, l. 13: which 'forcing data', please specify

**Reply:**

Forcing data here are prescribed atmospheric levels of greenhouse gases (GHGs). Details are given in Mitchell et al (2017).

This is specified in the new section describing the set-up of the experiments.

10) p. 21, l. 31: please give evidence that after 45 years a new equilibrium is indeed reached

**Reply:**

Below is a figure which shows the development of global temperature in the SO

simulations for present day 2005-2016 (SO-PD, black), the 1.5 (SO-15, blue), and the 2.0 (SO-20, red) degrees warmer world than pre-industrial. The numbering of years are arbitrary. All time series show considerable auto-correlation and multi-decadal variability. The weak, apparent trends from year 2161 to 2250 are not significant when data are sub-sampled based on the effective number of independent observations.

11) p. 22, l. 6: What is meant with 'single projections'?

**Reply:**

Single projections means that there is no ensemble of projections (e.g. with ensemble members starting from different initial states), but only one single estimate. We propose to remove "single".

We do not use the term single projections in the revised manuscript or in the Supplement.

**12) p. 22, l. 7:**

Table 7 does not show that the temperature targets (1.5 and 2.0 degrees above pre-industrial) are hit, but only show the temperatures relative to PD. What are the temperature relative to preindustrial?

**Reply:**

This is a consequence of the HAPPI-design (Mitchell et al, 2017), for which calculations are based on present-day (PD; 2006 – 2015) which is estimated independently at 0.8 o C above the pre-industrial. Since neither the SO- nor the AMIP-experiments are run explicitly for the pre-industrial situation, we will not present the pre-industrial numbers for these two sets of experiments. For the fully coupled experiments, however, we will add numbers for the pre-industrial situation.

The focus of the manuscript is on the changes in the 1.5 and 2.0 warming experiments relative to the PD climate as in the HAPPI project. This now clearly

stated several times in the revised manuscript. The PD climate is estimated to be 0.8 degrees warmer than the pre-industrial in the AMIP experiments. This is also now clearly stated in the manuscript. We show the time-evolution of the near-surface temperature response in the coupled warming experiments relative to pre-industrial conditions in Figure 2.

13)

As noted above, I found section 5.2 really hard to follow. What is particularly confusing is p. 22, I. 25-34. I think the sea ice area in the AMIP runs are compared to observations.

Questions I have here are:

1) why is this comparison not presented earlier,

2) why are there any differences if the prescribed data is based on observations?

3) why is sea ice extent in Table 4 compared with sea ice area in Table 8?

**Reply:**

There are obvious reasons for a reader to be confused here. In Table 4, sea-ice extent is presented, and not sea-ice area. In winter, the difference between area and extent (area being smaller than extent) may not be considerable, but still the comparison is not accurate. Nevertheless, we believe that the comparison indicates that there are some errors in the SO-model and probably also in the fully coupled NorESM1-Happi. The text in the paragraph will be reformulated and its emphasis will be reduced. This should have been done during the quality check before submission, and for this we apologize.

The text in the paragraph has been reformulated. Table 4 (now Table S5) has been moved to the Supplement and we no longer compare these results to Table 8 (now Table 4).

**Anonymous Referee #2**

Received and published: 30 January 2018

The authors look at a range of different versions of the NorESM1 model, and consider how those models hold up against reanalysis. They consider changes in many modes of variability, specifically related to key regional changes. Overall the paper was not what I

expected, from the title I expected the paper would be about NorESM1-Happi, Paris Agreement and Arctic Amplification. Very little of this was even mentioned until Figure 17!

**As it stands the paper is a model description/validation paper, and a science paper. The science is completely lost due to the first part. Due to this, and a number of other major concerns, I recommend substantial corrections.**

**Major concerns**

**1.**

In my view, the paper needs to be split into two. A paper focussing on the Arctic Amplification differences under Paris would be very welcome. So one suggestion is to put everything up to Figure 17 in online material, and just start the paper from there. Any reader that comes to this paper due to the title will be otherwise be completely lost in details of various models, and it will not be a productive read for them. More material would be needed for the science part though (see comments below). Alternatively, you could make this a model development only paper.

**Reply:**

As mentioned in the response to Referee #1, we agree that for the publication in ESD, the paper should be considerably restructured, and we propose to condense and move considerable parts of the sections that address pure validation of the NorESM1- Happi model, into a "supplementary material". In the main text, we will only keep those parts, which are directly relevant for the discussion of the difference between a 1.5 and a 2.0 degrees warmer world than pre-industrial, emphasizing the polar amplification of the temperature response.

Tentatively, we consider moving the following to supplementary material:

Tables 1, 2, 3, 4, and 6

Figures 1, 3, 4, 5, 6, 7, 8, 10, 11, 12, 15, and 16.

(Some of these figures will also be considered removed.).

There are also some numbers in the present main text that probably will be summarized in a new table. The text belonging to these items, which now fills up several pages in the manuscript will be compressed into one sub-section of the model description chapter. The text in the supplementary material will predominantly be written as extended Table headings and Figure legends.

What may remain in the main text are the discussions on Arctic amplification and the design of external forcing for reaching the given temperature targets with the fully

coupled NorESM1-Happi model. Furthermore, we will keep the discussion on the representation of sea-ice (Figure 9), extratropical cyclone activity (Figure 13) and blocking (Figure 14). Thus, the paper will be shorter and much more focused, while still documenting important aspects of NorESM1-Happi as a valid global climate model in the supplementary material.

We have shortened and moved the part of the paper concerned with the the validation of NorESM1-Happi against NorESM1-M, observations and reanalysis a Supplement. The paper is now focused on the role of ocean and sea-ice feedbacks under global warming of 1.5 and 2.0 degrees. We have expanded our analysis to include analysis of meridional temperature gradients and storm tracks at different levels. We have also added fully coupled simulations of the two warming targets and now present results from these simulations alongside results from the AMIP and SO simulations. The changes are listed in more detail under the reply to reviewer 1's overall response.

**2.**

The title makes it seem that NorESM1-Happi is the main model here, but actually it is not, the SO version is used the most, and the –M and –AMIP versions are used equally as much. I often got confused about which one was being used, as the paper jumped around a fair bit. It was not until half way through that I realised that the Happi version of the model did not have prescribed SSTs (as HAPPI is synonymous with prescribed SSTs).

**Reply:**

This confusion should be considerably reduced after the proposed paper reorganization. The definition of model versions (the AMIP, the slab-ocean (SO) and the fully coupled NorESM1-Happi) will be made already in the introduction. The NorESM1-M model is the CMIP5-version published in 2013, and is used only for documenting improvements in the NorESM1-Happi, and will thus be used for comparison in the supplementary material, and will not be prominent in the main text.

We consider to possibly use the name NorESM1-HappiCPL for the fully coupled model version, while NorESM1-HappiSO will be kept for the slab ocean version and NorESM1-HappiAMIP for the AGCM version.

This should be considerably less confusing in the revised manuscript, as we refer to the different model versions in a more consistent way. Also all the results for NorESM1-M are now in the Supplement, so the main text is now only concerned with the fully coupled, SO and AMIP versions of NorESM1-Happi. We have added a new section describing the set-up of the warming experiments with the different model versions, so it should be clear that the AMIP version has fixed SSTs and sea ice

**3.**

I was hoping to see more of a connection to Arctic Amplification here. Such as more of a

focus on latitude temperature gradients, changes in wave characteristics associated with this, and then relating this to blocking etc. This link was missing, and AA just seemed to be a 'hot topic' term. The authors should look at the recent work by Screen on this topic.

**Reply:**

It is mentioned in the introduction (p.3, I.26-20) that the intention of this paper's discussion of Arctic amplification is not to specifically address the potential impacts on planetary wave amplitudes and weather persistence.

We agree to extend the list of references to include publications by Screen, and we will mention the recently approved Polar Amplification MIP for CMIP6 (to which the NorESM-group plans to contribute). We should also mention that Figure 21 actually shows that there are more statistically significant changes in blocking occurrence in the NH in the slab ocean experiments than in the AMIP experiments. The AMIP experiment is not designed to produce a realistic Arctic amplification due to the way the sea-ice is prescribed (see our reply to pt. 4 of referee#1).

However, we disagree with the referee's statement that "AA just seemed to be a 'hot topic' term". Arctic amplification of the global warming is well established as a temperature signal, which deserves attention in its own right. Indeed, when discussing the difference between a "1.5 degree world" and a "2.0 degree world", this may prove to be the difference between an ice-free summer Arctic or not (e.g. Sanderson et al, 2017; and the results shown in our Figures 22 and 23), because of the amplified temperature response in the Arctic.

We have strengthened the results sections and now consider results from 1.5 degree and 2.0 degree warming experiments with AMIP, SO, and fully-coupled versions of the model. Our focus is on how ocean and sea-ice feedbacks in the SO and fully-coupled models affect climate response under 1.5 degree and 2.0 degree warming, i.e. how the AMIP experiments differ from the ones with ocean models. We have changed the title to reflect this.

We have added results showing changes in the upper-level and lower-level equator-to-pole temperature gradients in the Northern Hemisphere along with changes in the baroclinic-wave activity at similar levels. These results are moreover compared to the multi-model analysis in Li et al. (2018). The results for blocking activity mostly inconclusive due lack of consistency between the different models, but also the low statistical significance of the changes.

We have added references to studies by Screen and co-authors.

**4.**

A more comprehensive analysis seems to have already been done by the authors, in Li et al, (https://www.earth-syst-dynam-discuss.net/esd-2017-107/). Can the authors highlight what

their study adds?

Reply: While Li et al is a multi-model study on the response of selected features of the large-scale atmospheric dynamics entirely based on the HAPPI protocol of AMIP experiments, our paper's intention is to focus on the Arctic amplification of surface temperatures, which is not well represented in the HAPPI experimental set-up.

As emphasized in our reply to pt. 4 of referee#1, the reason for this is that some properties of the prescribed sea-ice in the AMIP experiments are not realistic. It is likely to misrepresent the Arctic surface temperature amplification, and therefore the differences between the 1.5 degree and the 2.0 degrees global warming in the Arctic, including sea-ice cover itself. There is a component of Arctic amplification in the HAPPI AMIP-experiments, but this is dominated by the prescribed SSTs and sea-ice concentrations in the 1.5 and 2.0 degrees warmer climate.

One consequence of this can be seen in the smaller response in NH extratropical storminess in our own AMIP results than in the results from both the slab ocean runs and the fully coupled NorESM1-Happi runs with RCP2.6 (Fig. 19). In pt.3 above, we have furthermore already mentioned the changes in NH blocking (Fig.21) and in Arctic sea-ice (Figs 22 and 23)

We have in the pipeline for the updated manuscript, longer simulations with the fully coupled model targeting the 1.5 degree and the 2.0 degrees warmer world.

We now compare our results to those in Li et al. (2018) both in the results sections and in the summary and disucssion. We also emphasize the difference between our study, which is focused on the NorESM1-Happi and now the AMIP experiments differ from similar experiments with active ocean components, and the study by Li et al., which is a multi-model study considering AMIP experiments from five different models (including the AMIP version NorESM1-Happi).

5.

I was very surprised by some of the differences between the SO model and the AMIP model. Surely the AMIP model will have smaller biases that the SO model (e.g. some Scaife papers could be referenced). It is not always clear that this is the case.

**Reply:**

(It would help if the referee#2 more specifically pointed to which results he/she is surprised to see.) The way we have calibrated and relaxed (i.e. bias corrected) the slab-ocean model for present-day conditions, we expect only small differences between the AMIP and SO model when comparing with e.g. re-analysed data for present-day conditions. This was also found for NH storminess and blocking (Figs. 18 and 20), and these results encouraged us to further employ the SO-model for the purpose of Arctic amplification on the target scenarios.

A more traditional way to calibrate the slab-ocean model (e.g. to study equilibrium climate sensitivity as mentioned in section 3.1), is likely to produce larger biases than in AMIP runs, where SSTs are prescribed from observationally based data.

The reviewer is correct, the AMIP model does have smaller biases than the SO model. This is for instance clear from the new figures showing the biases in the near-surface temperature (Figure 5) and baroclinic wave activity (Figure 11) and we point this out in the revised text.

At this stage, I am not sure minor comments are useful.

**The "NorESM1-Happi" used for evaluating the role of ocean and sea-ice feedbacks under global warming of 1.5 °C and 2 °C**

Lise S. Graff1, Trond Iversen1,2, Ingo Bethke3, Jens B. Debernard1, Øyvind Seland1, Mats Bentsen3, Alf Kirkevåg1, Camille Li4, Dirk J. L. Olivié1

[revised manuscript text omitted]
year averaged SSText determined by
the Operational Sea Surface</li> <li>Temperature and Sea Ice Analysis</li> <li>(OSTIA) for 2005–2016 (Donlon et al.,
2012), thus reducing SST biases. No
restoring of sea ice.</li>  | 150               |
| SO-15                | Equilibrium climate change for
an global surface air
temperature response of 0.7 K
above PD. | Forcing agents as in AMIP-15.
$Q_f$ calculated as for SO-PD by adding
the CPL-15–CPL-PD increments to the
OSTIA (2005–2016) climatology.                                                                                                                                                                  | 150               |
| SO-20                | Equilibrium climate change for
an global surface air
temperature response of 1.2 K
above PD. | Forcing agents as in AMIP-20.
$Q_f$ calculated as for SO-15 using the
CPL-20–CPL-PD increments.                                                                                                                                                                                                              | 150               |

Table 3: the NH and SH polar amplification factor (NH-PAF and SH-PAF) and global-mean near-surface temperature ( $T_{as}$ ) in the PD experiments and differences associated with 1.5 K warming, 2.0 K warming, and the 0.5 K difference for NorESM1-Happi, NorESM1-HappiSO and NorESM1-HappiAMIP. PAF is defined as  $\Delta T_{Polar}/\Delta T_{Global}$ , where T is the near-surface temperature, and the Global and Polar (poleward of 60 °) subscripts indicate the averaging region.

|                                   | Period or Difference | NH-PAF             | SH-PAF             | T as
K |
|-----------------------------------|----------------------|--------------------|--------------------|----------------------|
| NorESM1-                          | AMIP-PD              |                    |                    | 287.30               |
| HappiAMIP                         | AMIP-15-AMIP-PD      | 2.34               | 1.62               | 0.71                 |
| 125×10                            | AMIP-20-AMIP-PD      | 2.17               | 1.35               | 1.20                 |
| years                             | AMIP-20-AMIP-15      | 1.93               | 0.95               | 0.49                 |
|                                   | SO-PD                |                    |                    | 287.1 <mark>3</mark> |
| NorESM1-
Happi <mark>SO</mark> | SO-15–SO-PD          | 2.98               | -0.04              | 0. <mark>56</mark>   |
| 90 years                          | SO-20–SO-PD          | 2. <mark>68</mark> | 0.30               | 1.02                 |
| ·                                 | SO-20–SO-15          | 2.2 <mark>9</mark> | 0.77               | 0.43                 |
|                                   | CPL-PD               |                    |                    | 286. <mark>72</mark> |
| NorESM1-
Happi                 | CPL-15-CPL-PD        | 3. <mark>60</mark> | 0.2 <mark>3</mark> | 0. <mark>69</mark>   |
| 90 years                          | CPL-20-CPL-PD        | 2.99               | 0.5 <mark>6</mark> | 1.15                 |
|                                   | CPL-20-CPL-15        | 2.81               | 1.06               | 0.46                 |

|                                       | Period or Difference | T Land
K | T Land
K | P Land
mm d -1 | P Land
mm d -1 | $\frac{\text{AREA}_{\text{Sealce}}^{\text{DJF}}}{10^6 \text{km}^2}$ | AREA JJA
10 6 km 2 |
|---------------------------------------|----------------------|------------------------|------------------------|-----------------------------------------|-----------------------------------------|---------------------------------------------------------------------|--------------------------------------------------------|
| NorESM1-
HappiAMIP
125×10 years | AMIP-PD              | 265.87                 | 292.62                 | 1.214                                   | 2.532                                   | 11.26                                                               | 5.81                                                   |
|                                       | AMIP-15-AMIP-PD      | +1.52                  | +0.84                  | +0.070                                  | +0.104                                  | -0.97                                                               | -0.54                                                  |
|                                       | AMIP-20-AMIP-PD      | +2.36                  | +1.65                  | +0.091                                  | +0.139                                  | -1.36                                                               | -0.86                                                  |
|                                       | AMIP-20-AMIP-15      | +0.83                  | +0.81                  | +0.021                                  | +0.035                                  | -0.39                                                               | -0.32                                                  |
| NorESM1-
HappiSO
90 years       | SO-PD                | 265.30                 | 292.44                 | 1.212                                   | 2.559                                   | 12.52                                                               | 5.48                                                   |
|                                       | SO-15–SO-PD          | +1.46                  | +1.12                  | +0.041                                  | +0.120                                  | -0.65                                                               | -0.86                                                  |
|                                       | SO-20–SO-PD          | +2.19                  | +1.87                  | +0.078                                  | +0.126                                  | -1.02                                                               | -1.41                                                  |
|                                       | SO-20–SO-15          | +0.73                  | +0.75                  | +0.036                                  | +0.006                                  | -0.36                                                               | -0.55                                                  |
| NorESM1-
Happi
90 years         | CPL-PD               | 265.33                 | 291.04                 | 1.248                                   | 2.337                                   | 12.51                                                               | 7.59                                                   |
|                                       | CPL-15-CPL-PD        | +1.44                  | +1.14                  | +0.048                                  | +0.136                                  | -1.41                                                               | -1.73                                                  |
|                                       | CPL-20-CPL-PD        | +2.41                  | +1.86                  | +0.073                                  | +0.161                                  | -1.93                                                               | -2.29                                                  |
|                                       | CPL-20-CPL-15        | +0.97                  | +0.71                  | +0.025                                  | +0.025                                  | -0.51                                                               | -0.56                                                  |

Table 4: Similar as Table 3, but for near-surface temperature over land, precipitation on land, and sea-ice area in the NH (20 °N–90 °N) during winter (DJF) and summer (JJA).

**Figures**

---

## Author Response (AR2)

**We want to thank the reviewers for their constructive suggestions and comments. We have responded to their points below. The reviewers comments are in black and our responses in indented blue italic text. A marked-up version of the manuscript showing the changes we have done follows after that.**

**Anonymous    Referee    #1**

This is a completely rewritten paper that compares the climate response to 1.5C and 2.0C global mean warming in uncoupled, slab-ocean and fully coupled simulations with the NorESM model. Compared to the previous version, the manuscript has greatly improved featuring 1) a more focussed approach, 2) the inclusion of fully coupled simulations and 3) a clearer motivation for using the slab-ocean version of the model. I appreciate the efforts that went into this. I do have, however, major and minor concerns that the authors should address before I can recommend publication of this manuscript.

**General:**

1) The paper focusses on 'the role of ocean and sea-ice feedbacks', implying that the difference in response between the model versions is only due to the fact that feedbacks are allowed in the slab ocean and fully coupled version and not allowed in the AMIP version. An important aspect that is not elaborated on however is the fact that the prescribed SST and SI fields in the AMIP runs are taken from the CMIP5 multi-model mean (more specifically: the HAPPI-mip protocol), and not from the coupled NorESM model. If the AMIP boundary conditions were taken from the coupled NorESM model instead, the difference between the coupled and uncoupled response and hence the 'role of ocean and sea-ice feedbacks' would presumably be substantially smaller (this could be tested). My guess is that the larger polar amplification in the coupled model is not because ocean and sea-ice feedback amplify the polar amplification, but simply because of the fact that the coupled NorESM has a larger polar amplification than the multi-model mean of the CMIP5 models (the boundary conditions of the AMIP model). This puts into question the authors' interpretation of results, in particular the importance of ocean and sea ice feedbacks in explaining the response difference between the different model versions. I don't think this is a show-stopper. Documenting the difference in response between AMIP, SO and fully coupled model versions is a usefull excersize, but I do have concerns regarding the attribution of this differences to coupled ocean and sea ice feedbacks.

*This is a valid point — many thanks for pointing it out! To assess whether the Arctic amplification in NorESM1-Happi is indeed larger than the CMIP5 multi-model mean (used for calculating SSTs for HAPPI), we have computed the polar amplification factor (PAF) for several RCPs, including RCP2.6, using the CMIP5 models that were used to create the SST increments for the AMIP runs, and the corresponding simulations from NorESM1-Happi. The results show that the PAF for NorESM1-Happi is indeed in the upper range of the CMIP5 responses. We have added a figure showing this (Figure 8 in the revised manuscript), along with discussion both in the results section and in the summary and discussion section.*

*It is now clearly stated that though they may be a contributing factor, the feedbacks are not the sole contributors to the differences between the AMIP and the SO and fully coupled simulations. We have toned down the focus on sea-ice and ocean feedbacks in the abstract, introduction, and summary and discussion. We have also changed the title from "The NorESM1-Happi used for evaluation the role of ocean and sea-ice feedbacks under global warming of 1.5℃ and 2.0℃" to "Arctic amplification under global warming of 1.5℃ and 2.0℃ in NorESM1-Happi".*

2) While I appreciate the addition of the fully coupled simulations, the authors have not included a description of how the scenarios for these simulations were constructed (section 2.2). How was it determined that the combination of RCP2.6 forcings and the adjusted CO2 evolution would result in global mean temperature stabilization? What was the physical reasoning behind these choices? Sanderson et al. (2017) constructed the scenario using an emulator, Sigmond et al. (2018) established stabilization by switching off all anthropogenic emissions, but how did you determine the scenario?

*The details of the fully coupled scenarios were determined through an iterative trial-and-error process. We have added more description of how the scenarios were constructed in the revised manuscript.*

3) I find the structure of sections 2-4 non-intuitive. I would make this one section with section 2.1 describing the model and section 2.2 describing the model versions.

*We have restructured sections 2–4. Now the description of the model comes first (including the slab-ocean version), followed by the section about the CMIP5 experiments and then the 1.5 K and 2.0 K warming experiments.*

**Other:**

4) There are still quite a number of typos, e.g. P. 1 , p. 25: increase --> increases, P. 1, p. 29: it-->is, p. 2:l. 10: is to presented --> is to be presented

*Sorry about this. We have corrected the mentioned typos and proofread the manuscript.*

5) l. 22: The combination of 'Compared to the AMIP runs' and 'relative to the present day climate' is confusing. Perhaps remove 'relative to the present day climate' ?

*This sentence has been rewritten.*

6) p. 2, l. 29, p. 6, l. 9 and p. 19 l. 14: An other relevant paper that should be cited here is doi: 10.1038/s41558-018-0124-y who performed 1.5C and 2.0C stabilized warming simulations with a coupled model, by switching off all anthropogenic emissions in a 'free-CO2' mode

*We are aware of this reference and already cited it a few times (p 17 l 14 and p 19 l 7 in the previous version of the manuscript). The reviewer is however correct in that it is relevant in other places as well and it has been added where suggested.*

7) p. 4, line 24: 'specific to our set-up': a bit confusing, this suggests that the points listed below this statement are specific to the NorESM model and hence differ from the standard HAPPI specifications, but I don't think that is meant by the authors

*We agree that this is confusing. The description of the set-up includes the treatment of the sea-ice thickness which is not a part of the HAPPI protocol, and is as such is specific to the set-up of the HAPPI experiments in the NorESM. To make this clearer, we have removed the "specific to our statement" statement and taken the part about sea-ice thickness out of the bullet list, so that the list only includes the standard HAPPI specifications. The part about sea-ice thickness now directly follows the list in a separate paragraph.*

8) p. 6, l. 3: please also include the warming relative to the preindustrial run.

*CPL-15 is 1.51 K warmer than pre-industrial conditions and CPL-20 is 1.97 K warmer. This is now stated in the revised manuscript.*

9) p. 6, l. 28: Here is should be noted why the authors did not use the AMIP-15 and AMIP-20 boundary conditions to calibrate the SO model. If they had chosen that, the difference between the AMIP and SO responses (and hence the assessment of the importance of atmosphere-ocean feedbacks) would be much smaller (see my comment #1). Also, on p. 10, l. 29-30 the authors state that 'The experiments with NorESM1-HappiSO are designed to be comparable to the NorESM1-HapppiAMIP experiments'. Based on this statement it would more sense to me to use the AMIP-PD, AMIP-15 and AMIP-20 fields as an input to the calibration.

*The increments could alternatively be taken from the AMIP experiments and we have attempted to do this, but this resulted in strong changes in the Hadley circulation and in the jets during winter and spring for reasons we do not fully understand. This behavior is not seen in the AMIP runs and might not be realistic. Therefore, we use SST increments from the fully coupled runs rather than from the AMIP runs to make sure that the increments are consistent with the climate response of the coupled system in the model. This is now stated in the revised manuscript.*

10) p. 9, l. 28: 'SST is (in this connection) the mixed-later temperature: I suggest changing the notation to something like T_mix. Also, SST_ext should be defined here, and not later in the paper.

*For consistency, we have changed the notation of the mixed-layer temperature to $T_{mix}$, and the corresponding external field to $T_{mixExt}$. However, we also note in the text that in this version of NorESM, the mixed-layer temperature and sea surface temperature are equal. Therefore, we can still use observed SST as an external field during the calibration phase. We have also moved the definition of so $T_{mixExt}$. It is now defined in the first paragraph following equation 1.*

11). section 4.1: I think find this a bit confusing, it would make more sense if in p. 10 l. 16 alpha is not set to 0 (is this a typo?).

*You are correct, this is a typo and has been corrected. We have added the missing constants in equation (1) such that alpha = 1 is the natural choice during calibration.*

12) p. 11, l. 9: It may be useful for the reader to include here an explanation for why the AMIP forcing agents are used in the SO runs, and not those used in the coupled runs (with the adjusted CO2 concentrations).

*The purpose of the SO runs is to have experiments where the sea ice is free to respond to the imposed changes, but that otherwise are as similar as possible to the AMIP experiments. Furthermore, while the SO-model runs are estimating differences between states in equilibrium, the coupled runs are evolving with time. Therefore, the forcings are as far as possible the same as for the AMIP experiments, with the exception that the SST increments are from the fully coupled run (see reply to comment 9).*

13) p. 12, l. 5: I'm not sure how the smaller 1.5K-PD and 2.0K-PD differences can be explained by a cold bias in PD? Shouldn't the cold bias in the PD cancel out in the response? The 1.5K, 2.0K and PD are all expected to suffer from a cold bias, correct? (the same applies to p. 14, l. 1-3)

*It is well established that the model has a tendency to produce a too cold climate compared to observations and re-analyses. This is also shown in the supplement, and we also relate this to underestimated cloudiness and the strong Atlantic Meridional Overturning Circulation, which efficiently transfers heat into the deep ocean, leaving less for atmospheric temperature increase. (See Tables S3 and S4 and Figures S5, S7 and S14 in the supplement, and Figures 3 and 4 in the main paper.) We have slightly expanded on this discussion in the paper.*

*Note that biases can vary between different climate states in the same model version, so one cannot simply assume that they will cancel out when computing the differences between the warmer climates and the present-day climate. A in-depth discussion of the potential state-dependence of the biases in the model is, however, beyond the scope of the study.*

14) Fig. 18: The observational estimates (presumably shown by 'solid black contours'), are unclear. They are hardly visible and it is not clear what the contour levels are. It's probably better to show the obseverations in separate panesl. Also, there seems to be something wrong with the colors in panels e, f, i, j, with positive (>+2.5%) responses all the way down to the UK.

*We agree that the observations were hard to see, and have adjusted the color scale for the model fields to better show the observations. Contour intervals for the observations are the same as for the model. We have not opted to show the observations in a separate panel, as we feel this would make it more difficult to compare the observations with the fields from the models. Also, by an incurie, the same observational month was shown in both March and September; this has been corrected.*

*The contour intervals in panels e–l was not consistent with the colorbar in panels e, f, i, and j, which resulted the figures indicating increased sea ice concentration in large areas. Now, this has been corrected and panels e–l all have the same contour intervals.*

15) Section 8: The last 2 column of table 4 should be discussed here (not in section 5). What are the observed mean values of Sea ice extent/area, and how do they compare to the CLP-PD and SO-PD values? Fig. 19 suggest to me that the interannual variability in the coupled model is biased low.

*We prefer to keep some of the discussion of table 4 in section 5 (now section 4), but we now also discuss it in section 8 (now section 7). We have also added observed mean values of sea-ice extent and a discussion of these in section 8. The interannual variability in the coupled model is lower than in the SO model, but this is likely due to the sea-ice cover being too thick in the former case. We now discuss this in section 8. We identified an error in Figure 19 (now Figure 20), this is now corrected and does not affect the interpretation of the results.*

16) p. 17, l. 3 and l. 11: In line 3 it is noted that the SO has too little sea ice. This suggest that the ice-free frequencies under warming would be overestimated, correct? If yes this should be noted/discussed in lines 12-14.

*The reviewer is correct, we now discuss this in connection with Figure 19.*

**Anonymous Referee #2**

The paper is much improved from before, and now reads in a coherent way, and is far less confusing, so I'd recommend minor corrections. The only thing I think needs clearing up is the key messages they are trying to get across. At the moment, I think the novelty of the paper is in comparing atmospheric dynamics across three different model setups, (which are an atmosphere-only, a slab-ocean, and a fully coupled model version), in the context of future projections. However, reading the paper, I believe the authors are putting more emphasise on the scientific understanding of these dynamical events, from single model experiments, rather than the differences between experimental setups. This is dangerous, and many of the features they report on (blocking etc), have significant biases, and there is a body of research suggesting that multi-model studies are needed to look at these.

In summary, I would suggest that the authors rework the abstract, discussions, and a little bit of the intro, to highlight more clearly that they are looking at how these different experimental setups can change the atmospheric dynamic responses in the model. I also suggest they play down a little bit the reported numbers, on, say the latitudinal gradients etc

*We agree with the reviewer and have toned down the focus on the ocean and sea-ice feedbacks in the abstract, introduction, and summary and discussion and in the title. We have added a paragraph to the summary and discussion emphasizing that differences between the AMIP experiments and the experiments with an active ocean model are affected by the experimental set-ups. We have also removed the numbers for the relative changes in the polar amplification factor from the abstract. (See also response to reviewer 1's general remark 1.)*

*The reviewer is correct that there are large biases in the blocking frequency, and we clearly state in several places that the results are generally inconclusive. While multi-model studies are absolutely needed, we still feel that it is worthwhile to report results from individual models.*

[revised manuscript text omitted]

---

## Author Response (AR3)

We want to thank the reviewer for the suggestion. We have responded below, with the reviewer comment is in black and our response in indented blue italic text.

**Anonymous    Referee    #1**

The authors have addressed my comments in a satisfactory manner. My only recommendation is to improve the abstract as follows. In the abstract the authors describe that factors like polar amplification and the response of the upper-level equator-to-pole temperature gradient are larger in the coupled simulations than in the AMIP simulations. By following up on my suggestion (much appreciated) the authors have identified a reason for this (p. 14), which should be briefly described in the abstract.

*We agree with the reviewer that a brief description should be included in the abstract and we have now added this.*

[revised manuscript text omitted]
 l̲i̲n̲e̲c̶u̶r̶v̶e̶) and the spread (b̲l̲u̲e̲ ̲s̲h̲a̲d̲i̲n̲g̲;̲ ± one standard deviation) computed over the number of available decades (9 for NorESM1-Happi and NorESM1-HappiSO, and 125 for NorESM1-HappiAMIP) for the default time periods given in Sect. 3. Blocking frequency from ERA-Interim is shown for the period 1986–2015 (dotted black ̲l̲i̲n̲e̲). The blocking events are identified using the the vTM index (Tibaldi and Molteni, 1990; Pelly and Hoskins, 2003), as in Iversen et al. (2013). It is based on the TM-index (Tibaldi and Molteni, 1990), which uses a persistent reversal of the meridional gradient of the 500-̲hPa geopotential height around the predefined central blocking latitude at 50-°N as an indicator for blocking. The reversal must be present at 7.5-° consecutive longitudes and persist for at least 5 days. In the vTM index the requirement of a predefined central blocking latitude is relaxed in order to reduce spurious detection (Pelly and Hoskins (2003). The central latitude is allowed to vary with longitude following the latitude of the maximum in the climatological storm track (using bandpass-filtered geopotential height at 500 hPa). To account for the seasonal cycle of the cyclone activity, the central latitude for a given month is calculated as the climatological 3-month moving average centred on that month. Units are % (a–f).

**Blocking frequency (%, vTM index)**

[Figure]

Figure 18: change in blocking frequency (solid black line with red and blue shading) in the 1.5 K experiment relative to PD (top three rows; a–f) and for the additional 0.5 K of warming (2.0 K–1.5 K; bottom three rows; g–l), shown along with the blocking climatology for the PD experiment (dotted black line). The fields are shown for NorESM1-Happi (a, b, g, h), NorESM1-HappiSO (c, d, i, j), and NorESM1-HappiAMIP (e, f, k, l) during DJF (left; a, c, e, g, i, k) and JJA (right; b, d, f, h, j, l) for the default periods given in Sect. 3. The asterisks along the x-axis indicate where the changes at that longitude are statistically significant at the 5 % level according to the Welch t-test. Note that the left y-axis is for the difference field and the right y-axis is for the climatology. Units are % (a–l).

[Figure]

Figure 19: NH monthly-mean sea-ice concentrations for PD (top; a–d), the 1.5 K warming relative to PD (second row; e–h), and the 0.5 K warming (bottom row; i–l) from NorESM1-Happi (first and third column; a, c, e, , i, k) and NorESM1-HappiSO (second and fourth column; b, d, f, h, j, l). Fields are shown for March (first and second column; a, b, e, f, i, j) and September (third and fourth column; c, d, g, h, k, l). The modeled concentrations  are from the default 90-year periods (Sect. 3.2) . The PD results (colors; top color bar) are shown together with observational estimates (OSI-SAF, 2017; solid black contours) from 2006–2015. Differences that are not statistically significant at the 5 % level according to the Mann-Whitney U test are marked with black dots. Units are % of ocean surface area (a–l).

[Figure]

Figure 20: The relative occurrence of NH monthly-mean sea-ice extent in September for observations (black bars; OSI-SAF, 2017), the PD experiments (blue bars), and the 1.5 K (green bars) and 2.0 K warming experiments (red bars) from NorESM1-Happi (a) and NorESM1-HappiSO. The sea-ice extent is binned in 1.0×106 km$^2$ increments. The observations are from 1996–2015 (20 values) in (a) and from 2005–2015 (11 values) in (b). The values from NorESM1-Happi and NorESM1-HappiSO are from the default 90-year periods (Sect. 3.2). Units are $10^6$ km$^2$ (a–b).